# Physical limits to biomechanical sensing in disordered fibre networks

Farzan Beroz[1,2], Louise M. Jawerth[3,4], Stefan Münster[3,5], David A. Weitz[4,5], Chase P. Broedersz[1,2,6] & Ned S. Wingreen[1,7]

Cells actively probe and respond to the stiffness of their surroundings. Since mechanosensory cells in connective tissue are surrounded by a disordered network of biopolymers, their *in vivo* mechanical environment can be extremely heterogeneous. Here we investigate how this heterogeneity impacts mechanosensing by modelling the cell as an idealized local stiffness sensor inside a disordered fibre network. For all types of networks we study, including experimentally-imaged collagen and fibrin architectures, we find that measurements applied at different points yield a strikingly broad range of local stiffnesses, spanning roughly two decades. We verify via simulations and scaling arguments that this broad range of local stiffnesses is a generic property of disordered fibre networks. Finally, we show that to obtain optimal, reliable estimates of global tissue stiffness, a cell must adjust its size, shape, and position to integrate multiple stiffness measurements over extended regions of space.

[1] Joseph Henry Laboratories of Physics, Princeton University, Princeton, New Jersey 08540, USA. [2] Arnold-Sommerfeld-Center for Theoretical Physics and Center for NanoScience, Ludwig-Maximilian University of Munich, Munich D-80333, Germany. [3] Department of Biological Physics, Max Planck Institute for the Physics of Complex Systems, Dresden 01187, Germany. [4] Department of Physics, Harvard University, Cambridge, Massachusetts 02138, USA. [5] School of Engineering and Applied Sciences, Harvard University, Cambridge, Massachusetts 02138, USA. [6] Lewis-Sigler Institute for Integrative Genomics, Princeton University, Princeton, New Jersey 08540, USA. [7] Department of Molecular Biology, Princeton University, Princeton, New Jersey 08540, USA. Correspondence and requests for materials should be addressed to C.P.B. (email: Broedersz@lmu.de) or to N.S.W. (email: wingreen@princeton.edu).

Mechanical cues can govern cellular behaviour in decisive ways[1,2]. The elastic properties of a cell's substrate have been shown to guide cell migration[3,4] and determine cell fate[5,6]. Eukaryotic cells, including fibroblasts, mesenchymal stem cells and cancer cells, attach to substrates via transmembrane protein complexes called focal adhesions, allowing the cell to sense stiffness[2,7–10]. Knockdown studies have established that this mechanosensing contributes both to motility and to the regulation of cell shape in three-dimensional *in vitro* systems that closely resemble *in vivo* cellular environments[10–16], where cells are surrounded by a loosely connected, disordered network of protein fibres, such as collagen or fibrin[17]. These biopolymers form a major structural component of the extracellular matrix (ECM), which serves as the physical scaffolding within which cells live and move. While it is clear that cells actively probe these extracellular networks, it remains unclear how ECM micromechanical properties impact mechanosensing.

Both *in vitro* experiments[18–22] and theory[23–27] have demonstrated that biopolymer networks exhibit rich macroscopic mechanical behaviour, depending sensitively on network connectivity. However, because the size of a typical cell is comparable to the pore size of the ECM[28], any mechanical information must be inferred by locally probing an extremely heterogeneous material. Although a few studies have begun to characterize this microscopic response[29,30], a theoretical understanding of how local mechanics are determined by the surrounding heterogeneous structure is still lacking.

How does the intrinsic heterogeneity of the ECM limit a cell's ability to learn about its global environment from purely local mechanical measurements? For the case of chemical sensing, consideration of the fundamental physical limits dates back to Berg and Purcell's consideration of noise due to the random arrival of diffusing particles[31–33]. Here we take a similar approach to explore the fundamental limits of mechanosensing, where, in contrast to chemical signals, the cues are static in time but distributed nonuniformly in space.

To quantify the physical limits of mechanosensing imposed by a cell's disordered environment, we investigate a simple model consisting of two components: the ECM as an elastic network that deforms in response to external forces, and the cell as an idealized measurement device that probes the stiffness of its surroundings. We found that experimentally-imaged collagen and fibrin networks and randomly-generated networks all yield a very broad range of modelled local stiffness responses, spanning roughly two decades. We observed that the broad distribution of local stiffnesses collapses onto a universal form for different fibre concentrations. We trace the origin of this universally broad range of stiffnesses to two intrinsic features of disordered networks: first, the local stiffness depends primarily on a small number of local fibres with consequently large variations, and second, these proximal fibres contribute to stiffness in a highly cooperative manner. Although we find that the local mechanics of the experimental networks are dominated by fibre bending, we show that pre-existing strain in the network can induce a transition to a stretching-dominated regime. Finally, we argue that to obtain accurate estimates of global ECM stiffness, cells must integrate multiple stiffness measurements over extended regions of space.

## Results

**Forming and modelling experimental network architectures.** Cells in connective tissue can glean information about their surroundings by pulling on the individual biopolymers of the ECM. However, on the short length scale of a typical cell, the measured mechanical response is sensitive to the intrinsic structural disorder of the ECM. To investigate the role of local mechanical disorder in a physiologically relevant system, we considered collagen networks, which form the primary structural component of the ECM[17]. We prepared a sample network by reconstituting fluorescently labelled collagen type-I monomers at a concentration of $c = 0.2\,\mathrm{mg\,ml}^{-1}$ and imaged its three-dimensional structure[34] (Fig. 1a, see Methods section for details). The network is loosely connected (with an average coordination number $z \simeq 2.9$) and highly heterogeneous at the cellular scale (with an average mesh size $\xi \simeq 6.5\,\mu\mathrm{m}$). This reconstituted collagen architecture (Fig. 1a) was used as an input to construct a mechanical network model where the fibres are treated as elastic beams that can bend and stretch (Fig. 2a,

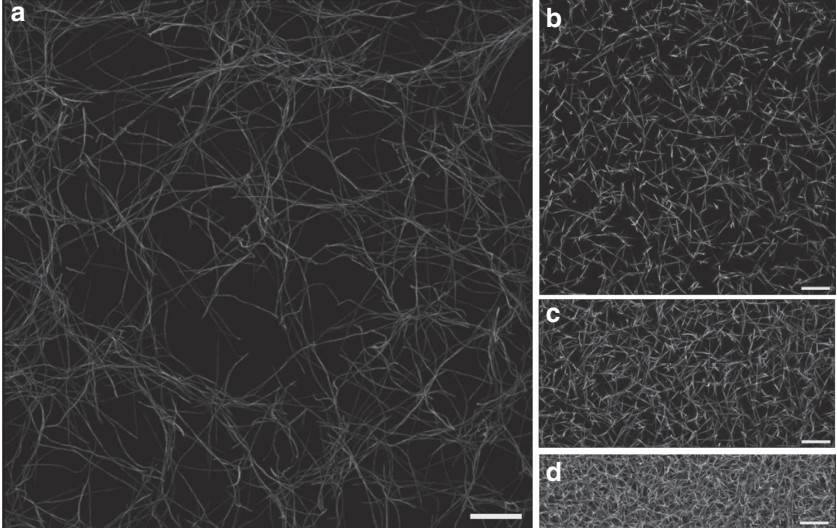

**Figure 1 | Experimental fibre networks.** (**a–d**) Reconstituted biopolymer networks considered throughout our analysis. Images show maximum intensity projection of three-dimensional confocal microscopy data obtained from polymerized (**a**) collagen type-I monomers at 0.2 mg ml$^{-1}$ and (**b**) fibrinogen monomers at 0.2 mg ml$^{-1}$, (**c**) 0.8 mg ml$^{-1}$, and (**d**) 1.6 mg ml$^{-1}$ (see Methods section for details of experimental procedure). Scale bars correspond to 25 μm.

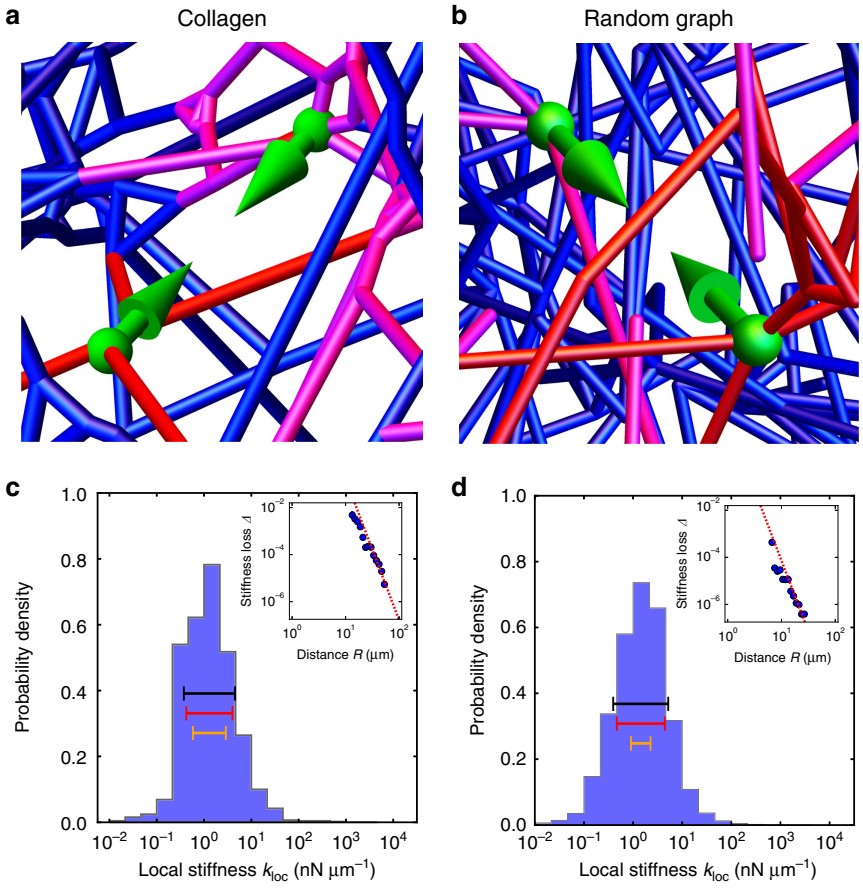

**Figure 2 | Force-dipole stiffness distribution.** (**a,b**) Examples of local stiffness sensing by force dipoles. Modelled deformation under stress from a local force dipole of length $d \sim 15 \, \mu m$ (green arrows) of (**a**) experimental collagen network and (**b**) RGG network. Magnitude of fibre deformations indicated by colour (small deformations, blue; large deformations, red). (**c,d**) Distribution of local stiffnesses $k_{loc}$ defined as the linear response of local deformation to a force dipole of length $d \sim 15 \, \mu m$ for (**c**) collagen network and (**d**) RGG network. Geometric s.d. of local stiffness $\sigma_{loc}$ indicated by bars (actual distribution, black; estimated distribution assuming strong locality, red; estimated distribution assuming weak locality, orange; see Supplementary Figs for details). Insets show stiffness loss $\Delta$, defined as the relative change in local stiffness $k_{loc}$ upon perturbing a network by removing a single fibre, versus distance $R$ of centre of removed fibre from the probe centre. For collagen, probe length $d < 10 \, \mu m$ and removed fibre length $\ell_{ij} < 10 \, \mu m$, and for RGG, probe length $d < 5 \, \mu m$ and removed fibre length $\ell_{ij} < 5 \, \mu m$. Error bars, defined as the s.d. of each data point divided by the square root of the number of samples averaged, are smaller than the size of data points. Dashed lines show asymptotic scaling from continuum theory, which predicts $\Delta \sim 1/R^{2D}$ for $R \gg d$ (see Supplementary Figs for details).

see Methods section for details). For simplicity, we modelled the stretching and bending of the beams, respectively, as springs and torsional springs connecting point-like vertices with stretching modulus $\mu$ and bending modulus $\kappa$. Throughout the main text, we take the same values of $\mu$ and $\kappa$ for all fibres of the network, and we include a bending interaction over each connected triplet of vertices. However, we also tested alternative mechanical models and found that our results are not significantly affected by these choices (see Supplementary Figs 3 and 4). Since biopolymers are expected to be much more pliable to bending than to stretching, we chose the stretching modulus such that $\kappa \ll \mu \xi^2$. The bending modulus $\kappa$ of the torsional springs was fitted to the experimental network using data from macroscopic rheology (see Methods section for details).

**ECM networks yield broad distributions of local stiffnesses.** To study the mechanical response of the network to local forces that might be applied by cells, we defined a 'local stiffness' $k_{loc}$ as the linear response of the displacements of two vertices to a dipole contractile force along the direction between the vertices[35]. We then calculated this local stiffness by numerically solving the equations of force balance for the network. Strikingly, we found that local stiffness measurements yield a very broad range of values, spanning up to roughly two decades (Fig. 2c). Because of the large range of stiffnesses, in what follows we characterize this variability in terms of the geometric standard deviation (s.d.):

$$\sigma_{loc} = e^{\sqrt{\langle (\log k_{loc} - \langle \log k_{loc} \rangle)^2 \rangle}} \qquad (1)$$

which we find for this collagen network to be $\sigma_{loc} = 0.54$.

To explore how the broad local stiffness distribution depends on protein type, we next considered a reconstituted fibrin network (Fig. 1b, Supplementary Fig. 1a), as fibrin constitutes the main structural component of blood clots[17]. We imaged a fibrin network prepared from a solution of fibrinogen at $c = 0.2 \, mg \, ml^{-1}$ and observed an average coordination number $z \simeq 2.7$ and an average mesh size $\xi \simeq 6.7 \, \mu m$ (see Methods section). We found that the fibrin network also has a broad distribution of local stiffnesses, with $\sigma_{loc} = 0.63$ very

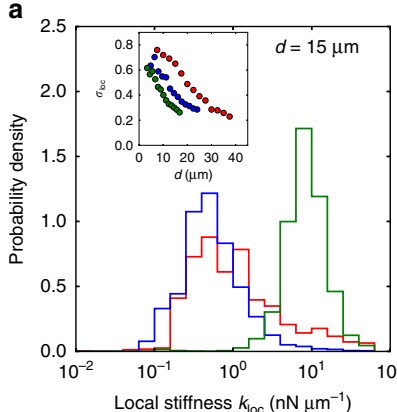
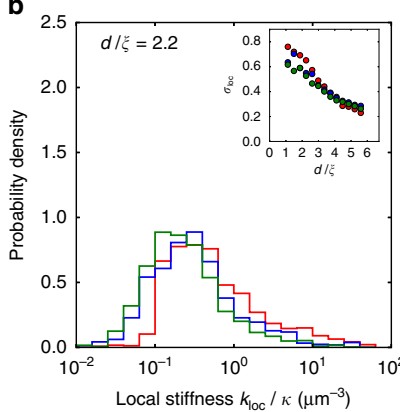

**Figure 3 | Varying concentration reveals universality of the local stiffness distribution.** (**a**) Distribution of local stiffnesses $k_{loc}$, at force dipole length $d \sim 15\,\mu m$, for fibrin networks with concentrations $c = 0.2\,\mathrm{mg\,ml^{-1}}$ (red), $c = 0.8\,\mathrm{mg\,ml^{-1}}$, (blue) and $c = 1.6\,\mathrm{mg\,ml^{-1}}$ (green). Inset shows geometric s.d. $\sigma_{loc}$ versus force dipole length $d$. (**b**) Distribution of local stiffnesses $k_{loc}/\kappa$ scaled by the bending modulus, obtained by fitting model parameters for each experimental network using data from macroscopic rheology, at scaled force dipole length $d/\xi = 2.2$, where $\xi$ is the mesh size, for fibrin networks with same concentrations as in **a**. Inset shows geometric s.d. $\sigma_{loc}$ versus scaled force dipole length $d/\xi$.

similar to that of the collagen network (Supplementary Fig. 1b). This similarity suggests that the local stiffness distribution may be determined primarily by network characteristics, such as connectivity and average mesh size, which were very similar for both types of networks.

To investigate how such network characteristics impact the local stiffness distribution of fibrin networks, we varied the fibrinogen concentration. We observed that, with increasing concentration, the mesh size and average fibre length both decrease (Fig. 1b,c and Supplementary Table 1). In addition, quantitatively comparing our models to macrorheological data yields a fitted bending modulus that increases with concentration (Supplementary Table 1). These denser networks of stiffer fibres also yielded broad local stiffness distributions, spanning at least two decades in stiffness. Interestingly, however, we found that $\sigma_{loc}$ became slightly smaller with each increase in concentration, for a fixed probe length (Fig. 3a).

To understand this reduction in $\sigma_{loc}$, we first note that the changes in the network features, including the mesh size and the fibre length distribution, were consistent with a simple size rescaling of the network (Supplementary Fig. 1c). Taken together, our results suggest that the width of the local stiffness distribution depends on the length scale over which the network is probed. To confirm this hypothesis, we varied the length of the cellular stiffness probe. In this case, we find that $\sigma_{loc}$ also decreases with increasing probe length for all concentrations of fibrin networks studied (inset of Fig. 3a).

The observed narrowing of the local stiffness distribution with monomer concentration or with probe length suggests that the stiffness distribution is controlled by the ratio of the probe length to an intrinsic length scale of the network. Indeed, upon rescaling the probe length $d$ by the mesh size $\xi$, we find that the geometric s.d. $\sigma_{loc}$ for networks of different monomer concentration collapse onto a single universal curve (inset of Fig. 3b). One possible explanation for this decrease in $\sigma_{loc}$ as a function of the ratio $d/\xi$ is that longer force dipoles effectively probe a larger region of the network, which may result in self-averaging. For long probes, however, $\sigma_{loc}$ begins to approach a constant value (inset of Fig. 3b). This asymptotic saturation occurs because for very long probes, the measured deformation is simply the sum of the deformations of two independent monopole probes that are still sensitive to network disorder.

**Random graph captures experimental networks' stiffness range.** To investigate the physical origins of the universally broad local stiffness distributions described above, we turned to idealized model networks. One way to generate model disordered networks consists of arranging vertices to lie on a regular lattice, such as a simple-cubic lattice or face-centred-cubic lattice (FCC) in three dimensions[25] (Supplementary Fig. 5a). Connecting these vertices randomly with a fixed probability results in a disordered lattice network. Such lattice networks have provided a useful starting point for characterizing the mechanical response of fibre networks in two dimensions[30]. Interestingly, although three-dimensional simple-cubic lattice and FCC networks also yield a broad range of local stiffnesses (Supplementary Figs 11f and 5b), the geometric s.d. of these distributions ($\sigma_{loc} = 0.29$ and 0.37 at $d/\xi = 2.2$, respectively) are considerably smaller than those of the experimental networks at the same value of $d/\xi$ (Fig. 2c; Supplementary Fig. 1b). Furthermore, the entire local stiffness distribution has almost no dependence on probe length (Supplementary Fig. 7d). A possible reason for these discrepancies is that the disordered lattice networks fail to capture important structural features of real networks at the scale of the cell, such as the fibre length distribution and the random positions of the vertices.

To generate a model network that better matches the structural features of the experimental networks, we begin by distributing vertices randomly throughout a volume. The density of these vertices (which determines the mesh size $\xi$) is set roughly equal to those of the experimental networks. Pairs of vertices are then connected according to a probability function that depends on the intervertex distance. We chose the probability function to be the simplest form that results in network features consistent with those of the experimental architectures (see Supplementary Methods). The resulting modelled networks are referred to as 'random geometric graphs' (RGGs; Fig. 2b). We computed the local stiffness distribution for the RGG network and found that the geometric s.d. is $\sigma_{loc} = 0.56$ at $d/\xi = 2.2$ (Fig. 2d), which is approximately equal to those of the experimental networks (Fig. 2c; Supplementary Fig. 1b). Furthermore, we find that both the experimental and RGG networks display an anomalously sensitive dependence of the geometric mean of local stiffness on probe length (Supplementary Fig. 6). Thus, in contrast to the lattice networks, the RGG network appears to quantitatively capture

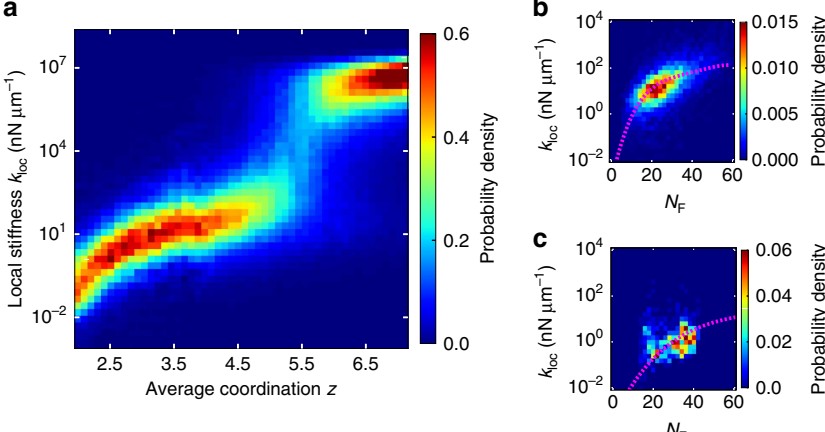

**Figure 4 | Collective effect of network structure on local stiffness.** (**a**) Distribution of local stiffnesses $k_{loc}$ for RGG network versus average coordination number of vertices, $z$, at force dipole length $d = 15\,\mu m$ and ratio of the bending modulus to the stretching modulus $\kappa/\mu = 10^{-5}\,\mu m^2$. (**b,c**) Joint distribution of local stiffness and number of local fibres $N_F$, defined as a weighted fraction of local bonds, with bond weight $= 1$ below a short-range cutoff $\tilde{\xi}$ and decaying as $1/R^{2D}$ beyond, where (**b**) $\tilde{\xi} = 1.5\xi$ for RGG network ($\xi \simeq 6.7\,\mu m$) and (**c**) $\tilde{\xi} = 2\xi$ for collagen network ($\xi \simeq 6.5\,\mu m$). Dashed lines show the macroscopic shear moduli of the networks as a function of average number of local fibres $\langle N_F \rangle$.

important local mechanical properties of the experimental networks.

**Experimental networks are in a bending-dominated regime.** What network features produce the broad width of the universal local stiffness distributions? Intuitively, more fibre-dense regions should be stiffer. For a fixed density of vertices, the fibre density is roughly proportional to the network connectivity, defined as the average coordination number $z$. We briefly review how the overall connectivity of a network affects its macroscopic mechanical properties[23,25].

For high values of $z$, the bulk elastic moduli of fibre networks are dominated by the stretching of fibres. As $z$ is lowered, the macroscopic response eventually undergoes a crossover to a bending-dominated response, as the network can be deformed without stretching fibres but not without bending them. As $z$ is further lowered, another elastic transition occurs when the network ultimately loses macroscopic rigidity. Over the whole range, the macroscopic response depends strongly and nonlinearly on the network connectivity, with the strongest dependence near the two elastic transitions.

To determine how the rapid scaling of the macroscopic response with connectivity manifests locally, we varied the average coordination number $z$ of modelled networks for a fixed ratio of the bending modulus $\kappa$ to the stretching modulus $\mu$ (with $\kappa \ll \mu\xi^2$, Fig. 4a, Supplementary Fig. 5c). At high connectivities, the entire local stiffness response is dominated by stretching interactions and scales with $\mu$. As the connectivity is lowered, the entire local stiffness distribution shifts to lower values. Specifically, the median stiffness decreases rapidly and nonlinearly with the average coordination $z$. Near the stretching–bending crossover, the local stiffness distribution becomes bimodal with the emergence of a subset of probes for which the measured stiffness scales with the bending modulus $\kappa$. Below the stretching–bending crossover, the number of stretch-dominated probes becomes negligible, and the local stiffness response enters a bending-dominated regime. Within this regime, the median stiffness resumes rapid decay. Finally, as the connectivity is brought below the rigidity transition, an increasing fraction of measurements yields zero stiffness as portions of the network become floppy. We thus find that the elastic transitions of the macroscopic response manifest locally as crossovers between stretch-dominated, bending-dominated, and zero-stiffness measurements. At these crossovers, the median stiffness decreases most rapidly, and the local stiffness distribution becomes bimodal, yielding a maximum in the geometric s.d.

The low connectivity of the collagen and fibrin networks suggests that they are situated in the bending-dominated regime. To verify this, we systematically varied the ratio of the bending modulus to the stretching modulus for these networks (Supplementary Fig. 2c,d). We found that the local stiffness measured by each individual probe scales with the bending modulus, confirming that we can account for the overall stiffening of the fibrin networks with concentration by rescaling local stiffness by the bending modulus. Indeed, upon plotting the distribution of rescaled local stiffnesses $\langle k_{loc} \rangle/\kappa$ for a fixed ratio of the probe length $d$ to the mesh size $\xi$, we find that the local stiffness distributions from all three fibrin networks collapse onto a single curve (Fig. 3b, Supplementary Fig. 1d). For all types of experimental networks we studied, no local stiffness measurements yielded zero stiffness or a finite stiffness that scales with the stretching modulus. The absence of such probes suggests that the experimental networks are far from both elastic crossovers, implicating a different source for the broad local stiffness distributions.

**Local stiffness depends only on proximal network structure.** A very broad distribution of local stiffnesses appears to be a generic feature of disordered fibre networks. What is the origin of this large variance in local stiffness measurements? To investigate how a local stiffness measurement depends on the surrounding network structure, we calculated the stiffness loss $\Delta$ upon removing a single fibre (Supplementary Fig. 10). Intuitively, removal of fibres that are more proximal to the stiffness probe should have a greater effect on $k_{loc}$, which suggests that, on average, the stiffness loss should decay as a function of the distance $R$ from the probe centre to the centre of the removed fibre. Indeed, for all types of networks we studied, we found that the average stiffness loss is consistent with $1/R^6$ decay (insets of Fig. 2c,d, Supplementary Figs 1b and 5b).

This apparent universality of the scaling of the stiffness loss suggests that the $1/R^6$ power-law decay should be calculable

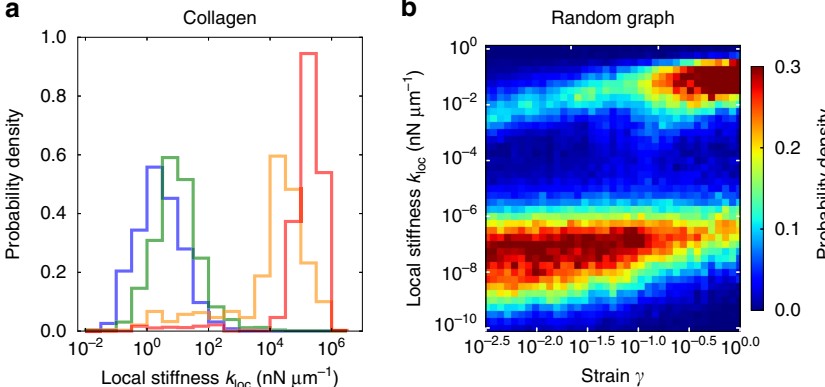

**Figure 5 | Effect of network prestress on local stiffness. (a)** Distribution of local stiffnesses $k_{loc}$, at force dipole length $d = 10\,\mu m$, for collagen network for a broad range of network strains (macroscopically uniform dilation $\gamma = 0$ blue, $\gamma = 0.01$ green, $\gamma = 0.05$ yellow, $\gamma = 0.2$ red). **(b)** Distribution of local stiffnesses $k_{loc}$ for RGG network versus network strain (average coordination number $z = 2.9$, force dipole length $d = 10\,\mu m$, $\kappa/\mu = 10^{-5}\,\mu m^2$).

within continuum elasticity theory. We therefore calculated the effect of removing a single fibre in the vicinity of a local stiffness probe for the case of a uniform lattice network (see Supplementary Fig. 10). The defect created by removing a fibre at a distance $R$ from the probe perturbs the dipole strain field induced by the probe. We can treat this perturbation as an additional dipole strain field originating from the defect, with a magnitude proportional to the initial strain in the removed fibre. Since both this initial strain and the consequent additional strain 'reflected' back to the probe decay as the strain field of a force dipole, that is, as $1/R^D$ (where $D$ is the dimension), the combined effect is an increase in the strain at the location of the probe $\sim 1/R^{2D}$.

The rapid $1/R^6$ decay of the stiffness loss due to fibre removal in three dimensions suggests that the local stiffness is largely determined by the network structure in the immediate vicinity of the probe. Within this small local region, all quantities are subject to large fluctuations; for example, since a small region typically contains a small number of fibres, the variance of fibre density will be large. The universal, rapid $1/R^{2D}$ decay implies that the mechanically relevant local structure will always be very local, with consequently large fluctuations.

**Broad stiffness range arises from local density fluctuations**. To quantify the dependence of local stiffness on the surrounding network structure, we considered the number of local fibres $N_F$, defined as the sum of fibres each weighted by a $1/R^6$-decaying function of its distance $R$ from the probe centre (see above and Supplementary Figs). We find that $k_{loc}$ and $N_F$ are well correlated for all types of networks studied, as shown in Fig. 4b,c. Importantly, $k_{loc}$ has a strong, nonlinear dependence on $N_F$. Specifically, the centre of the marginal distribution of local stiffnesses at fixed $N_F$ increases more rapidly than linearly with $N_F$. This indicates that local fibres influence local stiffness in a highly cooperative manner, that is, combining multiple fibres typically results in much larger than additive changes to local stiffness.

To estimate the geometric s.d. of local stiffness, we must account for these cooperative effects among fibres. We first note that the scaling of the median local stiffness with $N_F$ is consistent with that of the macroscopic shear modulus for all types of networks we studied (insets of Supplementary Figs 11d–f and 12d–f). This suggests that much of the broad width of the stiffness distribution can be accounted for by the large variations in the local fibre density taken together with the

strong, nonlinear dependence of the macroscopic shear modulus on overall fibre density. To test this notion, we estimated the distribution of local stiffnesses by taking $N_F$ transformed by the functional dependence of the macroscopic shear modulus, $G(N_F)$, where the modulus $G$ is that of a macroscopic network with an average number of local fibres given by $\langle N_F \rangle$. Upon accounting for the strong collective effects of fibre density in this manner, we found that the geometric s.d. of $G(N_F)$ provides a very good estimate for the actual, observed geometric s.d. of local stiffness (Fig. 2c,d; Supplementary Figs 1b, 5b, 11d–f and 12d–f). Consequently, the estimator correctly predicts the relative differences in the geometric s.d. for the different types of networks, including the observation that the stiffness distributions for the experimental and RGG networks are much broader than for the disordered lattice networks.

Our prediction of the width of stiffness distributions is not sensitive to the particular form of the weighting function, provided the decay is rapid enough. For instance, a hard cutoff at twice the mesh size captures a majority of the broad width for the FCC network (Supplementary Fig. 11e). However, a less rapidly decaying estimator includes a larger number of fibres in the local structure, and consequently the variance in fibre density is smaller. Thus, by comparison, an estimator with weights that decay as $1/R^3$ yields a smaller, much poorer estimate of the width of the stiffness distribution (Fig. 2c,d; Supplementary Figs 1b, 5b, 11d–f and 12d–f).

**Pre-existing network strain can induce an elastic transition**. Throughout our analysis, we have focussed on the local stiffness response of networks in an initially unstressed reference state. However, cells *in vivo* often encounter networks that are already in a deformed or stressed state. For example, the fibrillar proteins of the ECM are surrounded by proteoglycans (PGs), which form a hydrated, gel-like substance[17]. This PG gel has been observed to maintain an osmotic pressure that can swell embedded fibre networks and stabilize them in a stressed state[37]. An alternative source of stress is cells themselves, which actively contract and transmit forces throughout the network[7,38]. The presence of such pre-existing stress, or 'prestress', is known to impact the mechanical response of fibre networks[39,40], even at modest macroscopic strains $\gamma \sim 0.1$.

To determine how prestress could impact local stiffness, we considered the local stiffness for the experimental collagen network and the modelled RGG network under fixed initial prestrain. For simplicity, we did not include PGs or contractile

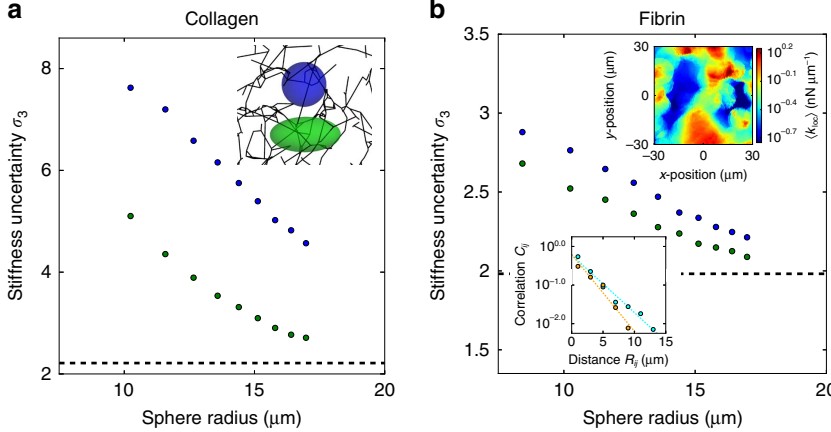

**Figure 6 | Minimum uncertainty of stiffness inference. (a,b)** Geometric s.d. $\sigma_3$ of the geometric mean of a random sample of $N = 3$ local stiffnesses measured by force dipoles whose centres lie within spheres (inset of **a**: blue) and prolate spheroids of equivalent volume (insets of **a**: green, aspect ratio 2:1; and red, aspect ratio 3:1) versus radius of spheres for (**a**) experimental collagen network and (**b**) experimental fibrin network. Dashed black lines shows the geometric s.d. $\sigma_{\mathrm{loc}}^{1/\sqrt{3}}$ of the geometric mean of $N = 3$ independent local stiffness measurements. Upper inset of **b** shows a two-dimensional slice of the fibrin network stiffness landscape, defined at each point as the geometric mean of local stiffnesses measured by all force dipoles whose centres lie within a sphere of radius 8.4 μm centred on the point. Lower inset of **b** shows the two-point geometric correlation $\rho_{ij}$ for local stiffness, defined as the covariance between log-local stiffnesses $\log k_{\mathrm{loc}}^{(i)}$ and $\log k_{\mathrm{loc}}^{(j)}$ divided by the logarithm of the geometric s.d. of local stiffness squared $(\log \sigma_{\mathrm{loc}})^2$, versus distance $R_{ij}$ for collagen (orange) and fibrin (cyan) networks.

cells but rather induced prestrain by imposing a macrosopically uniform dilation (see Supplementary Figs and Supplementary Methods). As before, we define the local stiffness as the linear response of two vertices to a contractile dipole force. These local stiffnesses are again computed by numerically solving the equations of force balance, obtained from the derivative of the Hamiltonian evaluated at the prestrained state (see Supplementary Figs).

For small prestrains, the distribution does not appear to change significantly, aside from a slight upward shift (Fig. 5a,b). However, for larger prestrains around $\geq 0.1$, the distribution becomes bimodal with a second, considerably stiffer peak. The emergence of this second peak occurs at prestrains consistent with the onset of nonlinear stiffening observed in previous studies of the macroscopic bulk modulus of fibre networks[39,40]. Here the width of the stiffer peak is broad, spanning roughly two decades. However, compared to the local stiffness distribution for zero prestrain, the geometric s.d. of this peak is lower by about a factor of two, similar to the value we observed for unstrained networks above the bending–stretching crossover (Fig. 4a). We confirmed that the entire local stiffness distribution is dominated by stretching interactions by systematically varying the ratio of the bending modulus to the stretching modulus for the collagen network at a fixed strain of $\gamma = 0.2$. We found that, for this value of strain, the entire distribution scales with the stretching modulus (Supplementary Fig. 4b). This demonstrates that prestress can induce an elastic transition in the local stiffness distribution, analogous to the crossover we observed upon varying connectivity. Thus we expect the results and analysis we performed for the unstrained networks to describe prestrained networks, although with the bending–stretching crossover shifted to lower values of connectivity with increasing prestrain.

**Extent of probed region sets accuracy of stiffness inference.**
The intrinsic heterogeneity of fibre networks presents a challenge for cells attempting to glean information from stiffness cues. That is, tissues with different global stiffness properties may have

significant overlap of their local stiffness distributions. In this case, a single local stiffness measurement would provide only a poor estimate of the global stiffness and identity of the tissue.

One possible strategy for cells to increase their inference accuracy is by averaging multiple stiffness measurements. With enough samples of local stiffness, this method would allow cells to reliably distinguish between global environments with different mechanical properties. We can visualize the local stiffness that a cell might infer at each point of the network by plotting the geometric mean stiffness of measurements obtained within a cell-sized sphere centred on that point (upper inset of Fig. 6b). The patterns in the stiffness landscape reflect the correlations of nearby stiffness measurements and extend over regions larger than typical cell sizes. More precisely, we find that the correlation function $C_{ij}$ between the log-local stiffnesses measured by two probes decays exponentially as a function of the distance $R_{ij}$ between their centres, with decay lengths around 3 μm (lower inset of Fig. 6b and inset of Supplementary Fig. 13a). These spatial correlations of local stiffness arise because nearby stiffness measurements depend on shared local structure. Since correlations reduce the effective number of independent samples, we expect less accurate global inference if stiffness measurements are made closer together in space.

To quantitatively study the reduction in accuracy of cellular mechanosensing due to spatial correlations, we modelled cell inference as an idealized sampling and averaging. To be concrete, we considered the sampled stiffness to be the geometric mean of a random sample of three probes whose centres are contained within prolate spheroids of varying volume and aspect ratio (inset of Fig. 6a). For all the types of networks we studied, the shape of the sampling region directly impacts the uncertainty of stiffness inference (defined as the geometric s.d. $\sigma_3$ of the inferred stiffness, Fig. 6 and Supplementary Fig. 13b). Specifically, the uncertainty is always reduced by increasing the volume of the sampling region as well as by increasing its aspect ratio. As the extent of the sampling region is increased, typical samples of stiffness become increasingly uncorrelated, and in the asymptotic limit, the uncertainty of the sampled stiffnesses approaches the geometric s.d. for independent samples.

## Discussion

Mechanical cues can guide cell motility and differentiation[3–6]. Indeed, cells are observed to reliably distinguish among two-dimensional homogeneous substrates with bulk stiffness similar to brain ($\sim 1\,kPa$), muscle ($\sim 10\,kPa$) and bone ($\sim 100\,kPa$)[5]. However, the three-dimensional *in vivo* mechanical environment at the cellular scale can be extremely heterogeneous[17,28]. Here we asked how this intrinsic heterogeneity impacts mechanosensing. Interestingly, we found that, within macroscopically homogeneous but locally disordered collagen and fibrin networks, local probes yield a very broad range of stiffnesses, spanning roughly two decades. This range is similar to the relative difference in the bulk stiffness of brain and bone. Moreover, the average measured stiffness is anomalously sensitive to probe length. We quantitatively captured these striking features of the experimental networks with modelled networks, which enabled us to elucidate their physical origins using a combination of simulations and scaling arguments.

We first established that the very broad range of local stiffnesses is a universal feature across network types and spans a wide range of connectivities, including multiple elastic regimes. We then traced the origin of this pervasive broad distribution to variations in local structure. Specifically, stiffness probes are primarily sensitive to a very small region of local fibres, and these privileged fibres contribute to the measured response in a highly cooperative manner. Finally, we found that the distribution of stiffnesses can be further broadened by tuning specific network features, including proximity to elastic transitions and geometrical disorder. While our results indicate that the experimental networks are poised squarely in a regime where the response is dominated by fibre bending, these networks include strong geometrical disorder, including randomly oriented fibre junctions and a polydisperse fibre length distribution. These structural features explain why the range of stiffnesses is larger for the experimental and RGG networks than for lattice-based networks.

The fibre concentrations we considered are typical for *in vitro* experiments performed on cell migration[10,36]. These values of concentration are chosen to be lower than the values *in vivo* because sparser networks allow for accurate imaging. Our observation of the universality of the local stiffness distribution provides a theoretical justification for these lower fibre concentration *in vitro* networks as models for *in vivo* networks.

Our 'ideal-mechanosensor' model does not address the internal mechanics of cells. Any internal noise in sensing or downstream signalling can only increase overall measurement uncertainty. There is evidence that cells can fix and thus regulate the relative uncertainty in measured stiffness by linearly modulating their applied stress to maintain a constant deformation[8]. Notwithstanding, we have shown that, even if cells sense nearly optimally, any single stiffness measurement is poorly informative of global tissue properties. This suggests that cells can benefit more from integrating the results of multiple stiffness measurements than from optimizing individual measurements, which may explain why some cells display more than a hundred focal adhesions[37]. Yet even this strategy has diminishing returns, because nearby stiffness measurements probe the same underlying local structure and are therefore correlated. To extract useful information, cells must spread their measurements over extended regions of space, either by moving or by extending their shape. The benefits of an elongated shape are twofold, since measurements on larger scales are both more accurate and less correlated. The biological relevance of this strategy is supported by the observation of highly polarized cells over five times longer than wide, including fibroblasts, mesenchymal stem cells, and cancer cells[5,13,41,42].

To better elucidate the physical origin of mechanical heterogeneity in fibre networks, we focussed on simple models that capture the essential features of the response of biological ECM. These models set a lower bound on the heterogeneity of the local mechanical response arising from intrinsic structural disorder. Biological ECM may contain additional sources of heterogeneity. Our simplified models approximate the thicknesses of fibres and crosslinking of fibres at junctions as uniform throughout the network and treat fibre deformations as purely elastic. To validate these simplifications, we tested alternative mechanical models that incorporate the most likely sources of additional heterogeneity: a distribution of bending moduli, different types of interactions at junctions, and plastic deformations (Supplementary Figs 3 and 4). For all these variations, we observed no significant difference in modelled stiffnesses on a case-by-case basis, which suggests that the large heterogeneity we observed from intrinsic structural disorder dominates over the most likely additional sources of heterogeneity and thereby provides a relevant estimate for the total heterogeneity.

Various other components are expected to contribute to the mechanical response that cells feel *in vivo*, including PGs and other cells. X-ray scattering suggests that PGs coat the fibres[43], effectively increasing their stiffness, while cells adhering to fibres serve as an additional elastic component in parallel with the network. Since biological networks are well below the stretching transition, we do not expect these passive stiffening effects to significantly alter the modelled response. However, in addition to their intrinsic stiffness, both PGs and cells can stabilize networks in a prestrained state. Our model indicates that this prestrain may induce an elastic transition and slightly narrow the local stiffness distribution. This occurs for macroscopic network strains larger than roughly 10%, which would be achievable for networks containing a dense population of cells. Thus cells may exploit prestrain to increase the accuracy of mechanical inference.

We have not considered the dynamics of the ECM, which may be influenced by the intrinsic viscosity of PGs on a short time scale of seconds[44] or matrix remodelling on scales of hours to days[45]. In contrast, cells have been observed to perform static measurements of stiffness that can take minutes to hours to develop stress[1]. In these cases, such short timescale and long timescale dynamics can be safely ignored. However, certain types of cellular protrusions have also been observed to oscillate with a constant period on the order of tens of seconds[46], which suggests that the dynamics of local stiffness are a promising direction for future study. Furthermore, while we have only considered isotropic networks, tissues can have aligned fibres due to remodelling and prestrain[12,47]. Our results for isotropic prestrain suggest that aligned networks should yield a narrower stiffness distribution when probed along a fixed direction. This may explain why the alignment of the ECM has been observed to coordinate with cell polarization to promote migration[48].

Finally, probing stiffness beyond linear response could provide cells with additional information. How hard would a cell need to pull to access the nonlinear regime? Nonlinear effects will certainly become significant when fibres begin to buckle[38]. For a single fibre equal in length to the mesh size, the Euler buckling thresholds are between 0.01 and 10 nN using the bending moduli we inferred for the collagen and fibrin networks. Forces of several nanonewtons are achievable by stronger cells, as well as by colonies that migrate collectively[49]. This suggests that the nonlinear regime is also an interesting direction for future study.

In summary, the disorder inherent in biological fibre networks places severe physical limits on the accuracy of cellular mechanosensing, suggesting that organisms must have evolved cellular-scale strategies to cope with this uncertainty *in vivo*.

Going forward, high-throughput gene deletion and mutation studies in conjunction with realistic patterned substrates[50] can help reveal the full array of internal components and pathways required for accurate mechanosensation.

## Methods

**Polymerization of collagen and fibrin networks.** We study network architectures obtained from experiment by analysing images of reconstituted fibre networks. Collagen networks were prepared and imaged as described previously[51]. Briefly, a sample of $c = 0.2 \, \mathrm{mg\,ml^{-1}}$ protein in $1\times$ DMEM with 25 mM HEPES was prepared from a solution of collagen type-I monomers with a small fraction of fluorescently labelled monomers at 4 °C. A fraction of monomers are labelled to avoid issues with fibre formation. However, these monomers are well mixed before the network is formed, which results in all fibres containing a sufficient portion of labelled monomers to allow for imaging. Therefore, our technique allows us to accurately image all fibres down to 200–500 nm resolution. Network formation was induced by neutralizing the sample's pH with 1 M NaOH and incubating it at room temperature for 4 h. The resulting network was imaged using a confocal microscope (Leica SP5, Wetzlar, Germany). We acquired a set of fluorescent images covering a three-dimensional volume that was representative of the network structure. To determine the fibre positions and connectivity, the image stacks were thresholded and subsequently skeletonized, which resulted in a one-voxel thick line representation of all fibres. We define a branch point as the junction between three or more fibres. The number of fibres that join at a branch point defines the branch point's connectivity $z$. In our mechanical model, the vertices are positioned at the branch points and end points of fibres. By counting the number of vertices within the network volume, we found a vertex density of $8.52 \times 10^{-4}$ vertices $\mu m^{-3}$.

Fibrin networks were prepared and imaged as described previously[51]. Briefly, solutions of human fibrinogen (Enzyme Research Labs, South Bend, IN) containing a small fraction of fluorescently labelled fibrinogen with protein concentrations of $c = 0.2$, $c = 0.8$, and $c = 1.6 \, \mathrm{mg\,ml^{-1}}$ were prepared in buffer (150 mM NaCl, 20 mM CaCl, 20 mM HEPES, pH 7.4). Network formation was induced by the addition of activated human alpha-thrombin (Enzyme Research Labs, South Bend, IN) to the fibrinogen solutions (final thrombin concentration: 0.1 mg ml$^{-1}$). After the samples were allowed to polymerize for 12 h, the resulting fibrin networks were subsequently imaged and the data were processed analogously to the collagen sample. The vertex densities were found to be $8.1 \times 10^{-4}$, $30 \times 10^{-4}$ and $90 \times 10^{-4}$ vertices $\mu m^{-3}$.

For the collagen and the fibrin networks, all proteins purchased exhibit purities $>90\%$ as demonstrated by the respective manufacturers via SDS–polyacrylamide gel electrophoresis. For the collagen network at $c = 0.2$ and fibrin networks at $c = 0.2$, $c = 0.8$, and $c = 1.6 \, \mathrm{mg\,ml^{-1}}$, we found average coordination numbers of $z \simeq 2.9$, $z \simeq 2.7$, $z \simeq 2.8$ and $z \simeq 2.9$, respectively. Although these average coordination numbers are low, the networks are all macroscopically rigid, as defined by a finite linear response to shear forces. Rheological measurements were performed on networks prepared under experimental conditions analogous to those of the imaged samples. For the collagen network, the measurements were done using an AR-G2 rheometer (TA instruments, New Castle, DE) equipped with a custom-made plastic plate of 25 mm and sealed with mineral oil[51]. The macroscopic mechanical response was measured by applying a small oscillatory strain and measuring the resulting stress, which resulted in a shear modulus of $G \simeq 0.3 \, \mathrm{Pa}$. The fibrin networks were measured with a 40 mm per 4° cone-plate geometry. Samples were sealed with mineral oil and allowed to polymerize for at least 12 h. The networks were then perturbed in the same manner as for the collagen network and the shear moduli were found to be $G \simeq 1$, 10 and 45 Pa, respectively.

Macroscopic rigidity requires macroscopic connectedness, that is, the presence of a spanning cluster of vertices. In our analysis, we identified all vertices that belong to the largest cluster using a union-find algorithm and removed all other vertices from the network. For both the collagen and the fibrin networks, we considered a spherical sample of radius $R_{\mathrm{sample}} = 65 \, \mu m$. Within this sample volume, the $N$ vertices are distributed approximately homogeneously on scales large compared to the mesh size $\xi$, defined as the radius of a sphere whose volume is equal to the average volume per vertex:

$$\xi = \frac{R_{\mathrm{sample}}}{N^{1/3}}. \tag{2}$$

For the experimental collagen network at $c = 0.2$ and fibrin networks at $c = 0.2$, $c = 0.8$, and $c = 1.6 \, \mathrm{mg\,ml^{-1}}$, we found $\xi \simeq 6.5$, 6.7, 4.3, and 3.0 μm, respectively. On scales comparable to $\xi$, the networks are intrinsically disordered due to heterogeneity in both the spatial positions of the vertices and the lengths of fibres. We approximate the lengths of the fibres as the distance between their end points. The lengths of the fibres are highly polydisperse, spanning over an order of magnitude in length with a substantial fraction of fibres that are very long compared to the average fibre length (Supplementary Figs 1c and 9a,b). The fibre length distribution peaks at small lengths (that is, below the average fibre length) and decays roughly monotonically beyond the peak.

**Mechanosensing model.** *Fibre network model.* We study the mechanical properties of crosslinked biopolymer networks, such as those of the ECM, using a fibre network model, which consists of a collection of linear elastic elements that are connected at point-like vertices. The elastic elements model the stretching and bending interactions of the constituent fibres, which we treat as elastic beams[23,25]. Furthermore, since biological collagen and fibrin networks often consist of branched architectures for which fibres are constrained to meet at fairly regular angles[26], we also consider the interaction required to deform the junctions at which multiple fibres are joined. The mechanical energy in the fibre network model is given by:

$$H_{\mathrm{lin}} = \frac{\mu}{2} \sum_{\langle ij \rangle} \frac{1}{\ell_{ij}} \left( \mathbf{u}_{ij} \cdot \hat{\mathbf{r}}_{ij} \right)^2 + \frac{\kappa}{2} \sum_{\langle ijk \rangle} \frac{1}{\ell_{ijk}} \left( \frac{\mathbf{u}_{jk} \times \hat{\mathbf{r}}_{jk}}{\ell_{jk}} - \frac{\mathbf{u}_{ij} \times \hat{\mathbf{r}}_{jk}}{\ell_{ij}} \right)^2, \tag{3}$$

where $\mathbf{u}_i$ is the deformation of vertex $i$ about its position in the undeformed reference state,

$$\mathbf{u}_{ij} = \mathbf{u}_j - \mathbf{u}_i \tag{4}$$

is the relative deformation of nodes $i$ and $j$, $\mu$ is the stretching modulus, $\kappa$ is the bending modulus and $\ell_{ij}$ is the length of the fibre $ij$ connecting vertices $i$ and $j$ in the unperturbed reference state (that is, the unstretched fibre length),

$$\ell_{ijk} = \left( \ell_{ij} + \ell_{jk} \right)/2 \tag{5}$$

is the average unstretched length of fibres $ij$ and $jk$ and $\hat{r}_{ij}$ is a unit vector that points along the direction of the fibre $ij$ connecting vertices $i$ and $j$ in the unperturbed reference state. The first sum corresponds to stretching interactions and is taken over all fibres $ij$. The second sum constrains angular deflections of all connected pairs of fibres $ij$ and $jk$ and thus provides a minimal model for both the energy due to bending interactions as well as the interactions provided by semiflexble junctions.

*Idealized measurement device.* The cell is modelled as an idealized stiffness-measuring device that exerts force on vertices of the network. The effect of the applied force $\mathbf{f}_i$ perturbs the mechanical energy as follows:

$$\delta H = - \sum_i \mathbf{f}_i \cdot \mathbf{u}_i, \tag{6}$$

where the index is summed over all vertices. We assume that the ECM behaves as a viscoelastic solid and that the forces applied by cells change slowly. Assuming forces applied by cells change slower than the timescale set by viscous damping, the network will be deformed in a quasistatic manner, that is, the network reaches mechanical equilibrium for a given force. In this case, the deformation is completely determined by minimizing the full mechanical energy:

$$\frac{\mathrm{d}}{\mathrm{d}\mathbf{u}_i}(H + \delta H) = 0. \tag{7}$$

Expressing this equation in terms of the deformations and the forces leads to the equations of force-balance in static equilibrium:

$$\sum_j \mathcal{D}_{ij} \mathbf{u}_j = \mathbf{f}_i, \tag{8}$$

where $\mathcal{D}_{ij}$ is the force-constant matrix of the unperturbed Hamiltonian:

$$\mathcal{D}_{ij} \equiv \frac{\partial H}{\partial \mathbf{u}_i \partial \mathbf{u}_j}. \tag{9}$$

We model a two-point measurement performed by a cell as a force dipole, which is defined as a contractile force exerted on vertices 1 and 2 separated by a vector $\mathbf{d} = d \, \hat{\mathbf{r}}_{12}$ in the undeformed reference state. The vector of equal and opposite forces may be represented using Kronecker delta notation as follows:

$$\mathbf{f}_i = f_0 (\delta_{i1} - \delta_{i2}) \hat{\mathbf{r}}_{12}, \tag{10}$$

where $f_0$ is the magnitude of the force applied to each vertex.

The local stiffness $k_{\mathrm{loc}}$ is defined as the linear response of the displacement of vertices 1 and 2 to the contractile force:

$$k_{\mathrm{loc}} \equiv \frac{-f_0}{\mathbf{u}_{12} \cdot \hat{\mathbf{r}}_{12}}. \tag{11}$$

**Numerical procedure.** To calculate the local stiffness $k_{\mathrm{loc}}$ defined above, we compute the deformations of the vertices $\mathbf{u}_i$ numerically by solving the equations of force-balance in static equilibrium. These equations do not have a well-defined solution for all possible configurations of the applied force because the force-constant matrix is singular, that is, it contains eigenvectors with vanishing eigenvalues or 'zero modes'. If the force of a local stiffness probe couples to a zero mode, the resulting deformations will diverge and the local stiffness is undefined in linear response. Intuitively, this corresponds to probes that act on unconstrained, dangling portions of the network.

To solve the equations of force-balance while dealing with the technical challenge provided by zero modes, we compute the generalized inverse of the force-constant matrix. The generalized inverse allows us to first check whether a solution exists for a given force perturbation and then to solve for the deformations by multiplying the force perturbation by the generalized inverse. Such an approach is efficient because it allows us to measure local stiffness over an entire network using only a single matrix multiplication operation per probe, which is

computationally inexpensive compared with solving the system of linear equations anew for each local stiffness probe.

We obtain the generalized inverse by first computing the singular value decomposition of the force-constant matrix:

$$\mathcal{D} \Rightarrow \mathcal{S}\mathcal{U}\mathcal{V}^T, \tag{12}$$

where $u$ and $v$ are orthogonal matrices and $S$ is a diagonal matrix consisting of singular values $\{s_1, s_2, \ldots, s_r\}$ equal in number to the rank $r$ of the dynamical matrix. From this, the generalized inverse may be computed:

$$\mathcal{D}^+ = \mathcal{V}\mathcal{S}^+\mathcal{U}^T, \tag{13}$$

where $S^+$ is a diagonal matrix consisting of the reciprocals of the singular values $\{\frac{1}{s_1}, \frac{1}{s_2}, \ldots, \frac{1}{s_r}\}$. The generalized inverse provides a simple way to check whether a solution exists. That is, if a given force vector satisfies the following equation:

$$\mathcal{D}\mathcal{D}^+ \mathbf{f} = \mathbf{f}, \tag{14}$$

then the force-balance equation has a solution. This guarantees that a given force configuration has no projection onto any zero mode. For well-defined force probes, which satisfy equation 14, the resulting deformation of each vertex is finite and given by:

$$\mathbf{u}_i = \sum_j \mathcal{D}_{ij}^+ \mathbf{f}_j, \tag{15}$$

from which the local stiffness may be computed.

**Data availability.** The experimental network architectures generated during and/or analysed during the current study, as well as the code used for the analysis in the current study, are available from the corresponding authors on reasonable request.

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

## Acknowledgements

We thank Yigal Meir, Jason Rocks, and Pierre Ronceray for insightful discussions. This work was supported in part by the National Science Foundation Grants DMR-1310266 (to L.M.J., S.M. and D.A.W.), the Harvard Materials Research and Engineering Center DMR-1420570 (to L.M.J, S.M. and D.A.W.), a Lewis-Sigler fellowship (to C.P.B.), the German Excellence Initiative via the program 'NanoSystems Initiative München' (NIM) and the Deutsche Forschungsgemeinschaft (DFG) via project B12 within the SFB-1032 (to C.P.B. and F.B.), National Science Foundation Grants PHY-1305525 and PHY-1066293 (to C.P.B., F.B and N.S.W.) and the hospitality of the Aspen Center for Physics (to C.P.B. and N.S.W.).

## Author contributions

C.P.B. and N.S.W. conceived the research. L.M.J, S.M. and D.A.W. contributed to the experimental design. L.M.J. and S.M. carried out all experiments. F.B. performed all simulations and analysis. F.B., L.M.J., S.M., C.P.B. and N.S.W. contributed to the manuscript preparation.

## Additional information

**Competing interests:** The authors declare no competing financial interests.

