## [Peer Review File · Nature Communications]

Reviewer #1 (Remarks to the Author):

The manuscript discusses an important and timely topic in mechanosensing. However, I would like the authors to address the following issues before I can arrive at a decision.

1. How realistically does mechanosensing in the experimental networks reconstituted from purified extracts of type-I Collagen or Fibrin mimic the extracellular matrix (which consist of several other types of collagen fibers, proteoglycans, hyaluronic acid etc as well) of live cells?
2. In live tissues, the mechanics of the ECM presumably is affected via interaction with cells—the authors should discuss how this ECM-cell mechanical feedback might alter their results qualitatively or quantitatively.
3. The authors calculated the local stiffness loss upon removal of a single fiber? Given the presence of spatial correlations in how fibers are lost in load bearing networks—for example, more fibers are likely to be lost (fail/buckle) if some fibers are already missing, I would like the authors to discuss how the result will be modified for actual physiological scenarios... and if it stays the same, why.
4. The scaling of the stiffness loss as $(1/R^2D)$ is an important result of the paper, and the calculation should be at least discussed briefly in the main manuscript, even if the details are shown in the supplementary information. Furthermore, given there are no restrictions of space in the Supplementary Information, I would suggest the authors briefly mathematically show the substitution and expansion mentioned in "Upon substituting the continuum elastic Green's tensor in place of the lattice Green's tensor and expanding the perturbation to third order in $1/R$, we find the stiffness loss scales as ...". Also, what is the rationale for doing the expansion to third order – is it the lowest order non-vanishing term?

Finally, the manuscript would benefit from a more focused and clear discussion of the main objectives/conclusions of the study earlier than in the Discussion section. The paper might benefit from a more focused title as well, but I'd leave it to the discretion of the authors.

Reviewer #2 (Remarks to the Author):

This clearly written manuscript presents a computational framework for modelling disordered fiber networks and simulating the effect of cellular contractile forces on local stiffness distribution. Cells in connective tissues actively probe and deform the extracellular matrix proteins surrounding them and how the structural properties of this fiber network determine the accuracy and range of mechanosensing has not been completely understood. The references are thorough and well-used and the introduction sets up the paper well. The presented modeling framework provides a straightforward route to investigate the impact of heterogeneity yet the conclusions are of interest to a broad audience. Nevertheless, there are a number of assumptions made to simplify the model, some of which require further clarification and strengthening. In addition, a detailed representation of the experimental characterization part must be provided as the parameters measured in these experiments play an important role in the construction of the models and evaluating the relevance of the simulation results. Overall, I think the paper deserves publication in Nature Communication if the following points could be addressed.

1. A major uncertainty for this reviewer and future readers involves the quantification of experimentally-imaged samples. Raw and processed images of fluorescently labeled collagen and fibrin networks must be provided to allow the readers evaluate the resolution and quality of the data. Does labeling a fraction of the whole sample with fluorescent probes provide an accurate picture of network connectivity? Were the positions and connectivity of the fiber network determined in a 3D space by correlating the data extracted from each confocal section, by

processing a 3D deconvoluted volume or by processing a projection of all images on a single plane? A representative illustration of experimental images showing how the mesh size and the length of the fibers were measured is also missing.

2. How does network connectivity depend on the concentration of collagen monomers and fibrinogen/thrombin? Do the selected composition values recapitulate the in vivo environment of connective tissues? If the values are completely off, the networks may operate at different elastic regimes, which would of course affect the global stiffness inference. The authors ignore remodeling of fibers by the cells for the sake of simplicity, which is acceptable. However, a discussion on how remodeling can change network connectivity and mesh size is critical to again put the results of this work in context. Previous work has shown that unconstrained cell-laden collagen or fibrin constructs shrink in size due to cellular contractility and this event must lead to a change in network connectivity. When they are constrained, then this leads to the formation of anisotropic networks with fiber alignment, another phenomenon that was not addressed in this work for the sake of simplicity. Building upon the argument on cell polarization initiated by ECM alignment, a comprehensive analysis on how the results of this study must be utilized to understand the behavior of cells in loosely or densely connected connective tissues with or without boundary conditions during formation and at the steady state would greatly enrich the discussion.

3. It is unclear how the presence of a junction point (branch point) is determined. The fibers may reside on top of each other and even touch each other but these interactions do not guarantee the existence of a linkage. Fibers can slide with respect to each other and the degree of crosslinking may play an important role especially for the local stiffness response. A similar assumption has been made again during the construction of random geometric graphs. Pairs of vertices were connected according to a probability function that depends on the inter-vertex distance. A justification of these points is missing.

4. The authors assume that the cells are distributed apart from each other. How does the contractility of a cell in close proximity of another would affect stiffness measurements? While elongated shapes may provide more accurate and less spatially correlated measurements in a loosely populated network, increasing the density of force dipoles will lead to crosstalk between cells due to fiber alignment, a phenomenon more emphasized in elongated shapes (AS Abilash et al, Biophysical J, 2014 and H Wang et al, Biophysical J, 2014). From another perspective, recent work has provided evidence for a collective durotaxis behavior in epithelial cells emerging from supracellular transmission of contractile forces (R Sunyer et al, Nature, 2016). Would a coupled collection of measurement devices provide an accurate estimate of global ECM stiffness for individual cells in the absence of multiple measurements over extended regions of space?

5. The title is too broad for the work presented. Biomechanical sensing is performed in a variety of media and a disordered fibrillar network is only one example.

Reviewer #3 (Remarks to the Author):

Mechanical sensing by cells has been a topic of interest for a while, and the paper by Baroz and co-workers examines this issue by quantifying the molecular architecture of reconstituted ECM networks, and by modeling heterogeneity in the mechanical response of the network. The overall question is reasonable, however, I find the quantitative issues addressed by the author are already studied quite extensively by Mackintosh and co-workers, and others in the fiber-network modeling area. The use of experimentally derived fiber structures appears to be new, but I'm uncertain whether the results adds to our understanding of biomechanics sensing. Some specific issues are described below.

1) As far as I know, cell mechanosensing has been mostly addressed in 2D experiments. The

classical papers in this area are all done on 2D substrates. 3D mechanosensing is less clear, there are multitudes of things that are different in 3D. Cells could respond differently because of pore-size variation that lead to nutrient diffusion differences, for example. There has been no definitive studies of how cells respond differently in 3D. There has been data on cells migrating in 3D, but results there are probably due to other network properties like pore-size.

2) Even the 2D results must be carefully interpreted. For example, in a 2012 Nat. Mat. paper, some claim that it is not the matrix mechanical stiffness, but the number of collagen ligands. These issues are complex, and it is not clear cells are indeed sensing stiffness. I'm not sure how this paper adds to this discussion.

3) The title of the paper is confusing at least to me. I am not clear what the authors mean by 'physical limits' to mechanical sensing. Is it because the local stiffness is variable, it presents a limit to how the cells can sense the ECM stiffness? If so, it does not appear that this paper addresses this issue. The variation in local stiffness is to be expected, given the heterogeneous nature of fiber networks.

4) As I mentioned, network moduli heterogeneity and how it depends on fiber structure and filament bending properties have been studied. I am not sure how this paper adds to this area. There is also a large field that work on constitutive modeling, using very sophisticated models that incorporate fiber geometrical properties and fiber mechanics. The actual problem is even more complicated, since the fiber themselves have heterogeneous bending properties, and variable bundle sizes. All of the parameters in Eq. S2 should have some distribution. I suspect real network exhibit even higher variable local stiffness.

5) The phase transition in Fig. 2 as a function of connectivity does seem interesting. Perhaps a theory can be developed there. But people working in the percolation area may already know about this.

6) The stiffness as a function of probe length does seem interesting. Here I would like to see more results. For example, as one varies collagen concentration, how does the effective network properties (Fig. 3) change?

7) When a different collagen or fibrin concentration is used to make the ECM, how do the results in the paper vary? There are also issues like how ECM cross linking is done experimentally. I would have like to see a little more studies on network properties as a function of fiber density.

Reviewers' comments:

Reviewer #1 (Remarks to the Author):

The manuscript discusses an important and timely topic in mechanosensing. However, I would like the authors to address the following issues before I can arrive at a decision.

1. How realistically does mechanosensing in the experimental networks reconstituted from purified extracts of type-I Collagen or Fibrin mimic the extracellular matrix (which consist of several other types of collagen fibers, proteoglycans, hyaluronic acid etc as well) of live cells?

Fibrillar proteins are the dominant component of the ECM by mass, and cells probe the mechanical properties of the ECM by adhering to the surfaces of fibers. Based on the universality of the local stiffness distribution (see revised section B of main text), we expect that our results do not depend strongly on the type of fibrillar protein, and can be broadly applied to all types of disordered, semi-flexible polymer networks. We therefore expect the reconstituted experimental networks to provide a realistic description of the mechanical cues provided by the ECM. However, the other components of the ECM are known to impact the bulk mechanical properties of tissues in a variety of ways. Proteoglycans (PGs) are the second most abundant proteins of the ECM. Proteoglycans, together with hyaluronans, form a gel-like substance that fills the interstitial space between the fibers, which can provide a variety of contributions to the mechanical response of the fibers:

- Proteoglycans coat the fibers to increase their effective diameter[Ref. 42], which would be expected to effectively increase the bending modulus of the fibers. We found that the effect of increasing the bending modulus is a uniform increase of the local stiffness distribution, since each value of local stiffness scales with the bending modulus. This scaling occurs for values of bending moduli that are well below the stretching moduli, which we expect to hold even for PG-coated collagen fibers.
- Proteoglycans may influence the dynamics of the ECM via their intrinsic viscous response that occurs on short time scales of seconds[Ref. 43]. Cells have been observed to perform measurements of stiffness on time scales that range from seconds to hours. We have focused on the static response, which should apply to a wide variety of cellular probes that build up stress over minutes to hours. However, certain types of cellular protrusions oscillate with a constant period on the order of tens of seconds[Ref. 45], which suggests that the dynamics of local stiffness are a promising direction for future study.
- Proteoglycans contain fixed negative charges and therefore can swell due to osmotic pressure. In principle, this pressure can maintain the

networks in a deformed state, depending on how the networks are formed. We modeled this effect by considering the mechanical response of networks with preexisting strain, or “prestrain” (see new section G of the main text). We found that modest prestrains shift the bending-to-stretching crossover to lower connectivities, which can induce an elastic transition to the stretching regime.

We have addressed how the bending moduli and the bending-to-stretching crossover impact the elastic response of fiber networks. We therefore expect that our central theoretical framework should also apply to more complex composite networks. However, a complete characterization of the mechanical response of such composite networks will certainly require further experimental and theoretical work that we plan to pursue in the future.

2. In live tissues, the mechanics of the ECM presumably is affected via interaction with cells - the authors should discuss how this ECM-cell mechanical feedback might alter their results qualitatively or quantitatively.

In addition to other proteins of the ECM, the mechanical response of fiber networks to cellular probes of stiffness can also be affected by the presence of other cells.

- Cells provide a passive contribution to the stiffness of the network via the elasticity of their bodies, which mainly derives from their actin cytoskeleton. By adhering to the fibers, they serve as additional elastic elements that contribute to the effective coordination number of the network. However, since biological networks are well below the bending-stretching transition, the population of cells would need to be extremely dense to raise the effective coordination number of the networks enough to induce a transition.
- Cells adhered to the network can actively contract to deform their surroundings. We modeled this effect as a uniform preexisting strain and found that even modest values of prestrain are sufficient to induce a bending-to-stretching transition. Since the local stiffness distribution is slightly narrower in the stretching-dominated regime, we expect that tissues with dense populations of cells can readily exploit this effect to allow for more accurate stiffness measurements. To present these intriguing new results, we added the additional section G to the main text.
- Cells can remodel the matrix over the course of days. On such long timescales, we can safely model local stiffness measurements as the static response of networks to cellular probes. However, although we have only considered isotropic networks, prestrained and remodeled tissues may have aligned fibers. Our results for networks with preexisting strain (see part G of main text) imply that such an alignment may yield a narrower

stiffness distribution when probed along a fixed direction, which may explain why the alignment of the ECM has been observed to coordinate with cell polarization to promote migration. This suggests that matrix remodeling is an interesting direction for future study. We now state this point more clearly in the augmented Discussion.

Our theoretical framework describes how the mechanics of the ECM are affected by isotropic changes of the effective connectivity and prestrain. A complete understanding of the mechanical response of anisotropic networks is beyond the scope of our current paper. However, such networks are a promising direction for future study.

3. The authors calculated the local stiffness loss upon removal of a single fiber? Given the presence of spatial correlations in how fibers are lost in load bearing networks - for example, more fibers are likely to be lost (fail/buckle) if some fibers are already missing, I would like the authors to discuss how the result will be modified for actual physiological scenarios... and if it stays the same, why.

Since the modifications to the stiffness loss are subtle, we separated the discussion into two sections. In section E, we calculated the stiffness loss upon removal of a single fiber to begin our theoretical investigation of how a local stiffness measurement depends on the surrounding structure. In the following section, we went on to discuss the effect of cooperative effects among fibers. We demonstrated that consideration of the stiffness loss alone is incapable of reproducing the scaling of the marginal distribution of local stiffnesses, i.e. the “stiffness fluctuations,” with the number of local fibers. We found that the combined effect of removing multiple fibers results in larger than additive changes to local stiffness. We went on to capture this inherently collective effect by considering the scaling of the macroscopic shear modulus with overall network connectivity. These results generalize the stiffness loss upon removal of a single fiber to the case of multiple fibers. We expect these results to apply to networks with structures that are initially isotropic and uniform. However, as the reviewer notes, physiological scenarios may result in spatial correlations among the positions of fibers, which suggests that networks with correlated structures are an interesting direction for future study.

4. The scaling of the stiffness loss as $(1/R^{2D})$ is an important result of the paper, and the calculation should be at least discussed briefly in the main manuscript, even if the details are shown in the supplementary information. Furthermore, given there are no restrictions of space in the Supplementary Information, I would suggest the authors briefly mathematically show the substitution and expansion mentioned in “Upon substituting the continuum elastic Green’s tensor in place of the lattice Green’s tensor and expanding the perturbation to third order in $1/R$, we find the stiffness loss scales as...”. Also, what is the rationale for doing the expansion to third order - is it the lowest

order non-vanishing term?

We thank the reviewer for highlighting the importance of this result. The calculation is now outlined in the second paragraph of section E in the main text. The reviewer brought up a valid point regarding the substitution and expansion. Previously, we had omitted a step regarding the mathematics of taking the continuum limit of our lattice calculation. We have now shown the additional step involved in this calculation. Furthermore, we provided an explicit calculation that revealed the complete orientational-dependence of the stiffness loss. Indeed, the expansion is done to third order since it is the lowest order non-vanishing term, which we now explicitly state in the discussion of the calculation.

Finally, the manuscript would benefit from a more focused and clear discussion of the main objectives/conclusions of the study earlier than in the Discussion section. The paper might benefit from a more focused title as well, but I'd leave it to the discretion of the authors.

We agree that a focused discussion of the main objectives and conclusions should appear earlier in the main text. To that end, we streamlined the Introduction to present our findings in a more concise and clear manner. Furthermore, we agree that the initial title was not focused enough, since biomechanical sensing occurs in contexts aside from disordered fiber networks. Therefore, we decided to change the title to "Physical limits to biomechanical sensing in disordered fiber networks."

Reviewer #2 (Remarks to the Author):

This clearly written manuscript presents a computational framework for modelling disordered fiber networks and simulating the effect of cellular contractile forces on local stiffness distribution. Cells in connective tissues actively probe and deform the extracellular matrix proteins surrounding them and how the structural properties of this fiber network determine the accuracy and range of mechanosensing has not been completely understood. The references are thorough and well-used and the introduction sets up the paper well. The presented modeling framework provides a straightforward route to investigate the impact of heterogeneity yet the conclusions are of interest to a broad audience. Nevertheless, there are a number of assumptions made to simplify the model, some of which require further clarification and strengthening. In addition, a detailed representation of the experimental characterization part must be provided as the parameters measured in these experiments play an important role in the construction of the models and evaluating the relevance of the simulation results. Overall, I think the paper deserves publication in Nature Communication if the following points could be addressed.

1. A major uncertainty for this reviewer and future readers involves the

quantification of experimentally-imaged samples. Raw and processed images of fluorescently labeled collagen and fibrin networks must be provided to allow the readers evaluate the resolution and quality of the data.

We agree with the referee that it would be helpful to provide both raw and processed images. In addition to the network architectures extracted from the processed data, we have now included images of the experimental samples we analyzed (see new Fig. 1).

Does labeling a fraction of the whole sample with fluorescent probes provide an accurate picture of network connectivity?

A fraction of monomers are labeled to avoid issues with fiber formation. However, these monomers are well-mixed before the network is formed, which results in all fibers containing a sufficient portion of labeled monomers to allow for imaging. Therefore, our technique allows us to accurately image all fibers down to 200-500 nm resolution. We have included this additional useful information to clarify Supplementary Note 1.

Were the positions and connectivity of the fiber network determined in a 3D space by correlating the data extracted from each confocal section, by processing a 3D deconvoluted volume or by processing a projection of all images on a single plane?

The positions are determined by correlating data from each confocal section. We then use an image processing algorithm to skeletonize the network architecture, which results in 1-voxel representations of the backbones of the fibers. We defined branch points as the positions at which more than two fiber segments meet. For our mechanical model, we then take the positions of vertices to be the branch points and end points of fiber segments. The vertices were then connected according to the connectivity of the skeletonized network. To better convey these important details, we modified the Supplementary Note 1 accordingly.

A representative illustration of experimental images showing how the mesh size and the length of the fibers were measured is also missing.

The mesh size and length of fibers were determined by analyzing the positions of the vertices, or the “branch points.” That is, for simplicity our mechanical model treats fibers as straight segments connecting point-like vertices. This additional detail is now included in Supplementary Note 1. Since the vertices are obtained from the processed images in our analysis, we found it clearer and more precise to explain the procedure using equations, rather than including pictorial representations. Specifically, we took the fiber lengths to be the distances between their end points (see text below Eq. S2). The mesh size is calculated from the inverse density of branch points, given by the first equation of Supplementary Note 1.

2. How does network connectivity depend on the concentration of collagen monomers and fibrinogen/thrombin?

To address this important concern, we studied the dependence of the network structure on the experimental concentration of fibrinogen. As the concentration was increased, we observed denser networks of thicker fibers. However, the network connectivity remained largely unchanged (see Supplementary Table 1 for a summary of the network properties). The analysis of the modeled response of these additional networks led to the finding that the local stiffness distribution is universal, which we now detail in the initial sections of the main text.

Do the selected composition values recapitulate the *in vivo* environment of connective tissues? If the values are completely off, the networks may operate at different elastic regimes, which would of course affect the global stiffness inference.

Most *in vivo* situations have concentrations that are somewhat above our experimental range. However, even for biological networks at these higher concentrations, the ratio of the bending modulus to the stretching modulus is still expected to be well below the bending-to-stretching transition. Therefore, we expect that our results are applicable to connective tissue. Furthermore, the values we use are typical for *in vitro* experiments used for cell migration studies.

The authors ignore remodeling of fibers by the cells for the sake of simplicity, which is acceptable. However, a discussion on how remodeling can change network connectivity and mesh size is critical to again put the results of this work in context. Previous work has shown that unconstrained cell-laden collagen or fibrin constructs shrink in size due to cellular contractility and this event must lead to a change in network connectivity. When they are constrained, then this leads to the formation of anisotropic networks with fiber alignment, another phenomenon that was not addressed in this work for the sake of simplicity. Building upon the argument on cell polarization initiated by ECM alignment, a comprehensive analysis on how the results of this study must be utilized to understand the behavior of cells in loosely or densely connected connective tissues with or without boundary conditions during formation and at the steady state would greatly enrich the discussion.

We briefly discussed matrix remodeling in our response to the second comment of Reviewer 1. The previous version of our manuscript addressed the effect of varying connectivity, but not the mesh size or the network alignment. Based on our study of networks at varying concentration, we found that the local stiffness distribution has a universal form that is controlled by the ratio of the probe length to the mesh size. Therefore, we expect that networks which shrink due to contraction will yield a slightly narrower local stiffness

distribution. To address the effect of alignment due to cellular contractility, we studied the local stiffness distribution for networks with preexisting strains large enough to nonlinearly deform the networks. We found that even modest values of such “prestrain” could induce an elastic transition, which resulted in a narrower local stiffness distribution. Taken together, these results suggest that cellular contractility could readily be exploited to increase the accuracy of stiffness inference. These results led to substantial modifications of the main text of our paper.

3. It is unclear how the presence of a junction point (branch point) is determined.

The branch points are determined by locations in the voxel-representation at which two distinct lines meet. We have included this additional useful detail in Supplementary Note 1.

The fibers may reside on top of each other and even touch each other but these interactions do not guarantee the existence of a linkage. Fibers can slide with respect to each other and the degree of crosslinking may play an important role especially for the local stiffness response.

Since our experimental technique cannot access the mechanical behavior of the junctions, we assumed that all junctions are rigid in our initial mechanical model. However, some types of crosslinkers allow fibers to rotate or slide freely. We tested an alternative model for crosslinking and found that choosing freely-hinging crosslinks at fourfold vertices (i.e. wherever two fibers may be crosslinked) does not significantly change our results. These additional modeling results are now included in Supplementary Note 3.

A similar assumption has been made again during the construction of random geometric graphs. Pairs of vertices were connected according to a probability function that depends on the inter-vertex distance. A justification of these points is missing.

For simplicity, we chose the probability function used to connect vertices in the random graph model to be a power-law with an exponentially-decaying cutoff. To determine this form, we tested a few different functions and chose the simplest version that provided a reasonable match to the network properties. This turned out to be a power-law with an exponent of 2 and an exponential function with a decay length fitted to the average fiber length. These choices allowed us to simultaneously fit the experimental fiber length distribution, mesh size, and average coordination number. We updated the main text to include additional clarification of these points.

4. The authors assume that the cells are distributed apart from each other. How does the contractility of a cell in close proximity of another would affect stiffness measurements? While elongated shapes may provide more accurate

and less spatially correlated measurements in a loosely populated network, increasing the density of force dipoles will lead to crosstalk between cells due to fiber alignment, a phenomenon more emphasized in elongated shapes (AS Abilash et al, Biophysical J, 2014 and H Wang et al, Biophysical J, 2014). From another perspective, recent work has provided evidence for a collective durotaxis behavior in epithelial cells emerging from supracellular transmission of contractile forces (R Sunyer et al, Nature, 2016). Would a coupled collection of measurement devices provide an accurate estimate of global ECM stiffness for individual cells in the absence of multiple measurements over extended regions of space?

We thank the reviewer for this helpful suggestion, which helped inspire additional modeling work on the effect of fiber alignment due to contractility. The crosstalk among different cells will certainly become relevant for forces that produce large nonlinear deformations of the network. To understand how contractility could impact local stiffness, we studied the local stiffness distribution for networks with uniform preexisting strain (see section G of the main text). We found that even modest values of such “prestrain” could induce an elastic transition. Therefore, we expect that dense populations of cells could certainly harness this effect to obtain more accurate estimates of global ECM stiffness. To put our findings in the right context, we now cite Ref. 46 (Wang, H., *et al.* 2014) in the augmented Discussion of our paper.

5. The title is too broad for the work presented. Biomechanical sensing is performed in a variety of media and a disordered fibrillar network is only one example.

We agree that the initial title was not focused enough. Therefore, we decided to change the title to “Physical limits to biomechanical sensing in disordered fiber networks.”

Reviewer #3 (Remarks to the Author):

Mechanical sensing by cells has been a topic of interest for a while, and the paper by Baroz and co-workers examines this issue by quantifying the molecular architecture of restituted ECM networks, and by modeling heterogeneity in the mechanical response of the network. The overall question is reasonable, however, I find the quantitative issues addressed by the author are already studied quite extensively by Mackintosh and co-workers, and others in the fiber-network modeling area.

We thank the reviewer for highlighting the interest of cellular mechanosensing. Many studies by MacKintosh and others in the field focussed on macroscopic network response. Here we ask what the mechanical response looks like from the perspective of the cell. We observed that this local response is broadly heterogeneous, in contrast to the macroscopic response. Furthermore, no previous studies have explored the physical origin of this broad mechanical

heterogeneity. To address this important gap in the understanding of local response, we performed substantial theoretical work that paves the way for a completely new fundamental understanding of local mechanical heterogeneity.

The use of experimentally derived fiber structures appears to be new, but I'm uncertain whether the results adds to our understanding of biomechanics sensing.

Our work presents the first theoretical model that captures quantitative features of the local stiffness distribution. Initially, we tried to capture the local mechanical properties using lattice-based networks, which have been argued to capture the macroscopic response[Ref. 23]. Importantly, however, we found that these lattice networks fail to capture essential features of the local response of experimentally-derived networks, including the anomalous stiffening and broadening of the local stiffness distribution as a function of probe length. Based on these observations, we proposed a simple network model (RGG), which quantitatively captures these striking features of the local stiffness. Thus, our results demonstrate an important shortcoming of the existing lattice models and provide a simple model to overcome these shortcomings. We went on to perform the first quantitative estimates of how the local stiffness landscape arises from the structural disorder of fiber networks, along with a new conceptual framework for modeling cellular inference. We employed this framework to reveal new strategies that cells may employ to obtain reliable estimates of local stiffness. We also provided the first estimates of the local stiffness distribution of strained networks, which yields a new insight into how populations of cells might work together to obtain more accurate estimates of stiffness. We believe that these results add to our understanding of biomechanical sensing.

Some specific issues are described below.

1) As far as I know, cell mechanosensing has been mostly addressed in 2D experiments. The classical papers in this area are all done on 2D substrates. 3D mechanosensing is less clear, there are multitudes of things that are different in 3D. Cells could respond differently because of pore-size variation that lead to nutrient diffusion differences, for example. There has been no definitive studies of how cells respond differently in 3D. There has been data on cells migrating in 3D, but results there are probably due to other network properties like pore-size.

There is strong evidence that stiffness influences cell behavior in 3D. Specifically, stiffness has been found to impact cell motility and morphology in a number of studies that employed realistic, three-dimensional networks to study cell behavior *in vitro*:

- The authors of Ref. 10 found that stiff fibers in 3D reconstituted

collagen networks can enable the formation of stable focal adhesions, consistent with results on 2D substrates.

- The authors of Ref. 11 found that the speed of cell migration through 3D Matrigel depends on the stiffness of the Matrigel.
- In Ref. 13, the authors demonstrated that the mechanosensory protein Vinculin is required for cell polarization, migration, and ECM remodeling in 3D collagen.
- The authors of Ref. 14 found that matrix stiffness drives epithelial-mesenchymal transition and tumor metastasis through a mechanotransduction pathway.
- In Ref. 15, the authors employed a microenvironment that allowed ECM stiffness to be controlled independently from composition and architecture. They found that *increasing ECM stiffness alone* resulted in a malignant phenotype in normal mammary epithelial cells.
- In Ref. 16, the authors showed that distinct ECM mechanosensing pathways control endothelial cell branching morphogenesis.

These studies demonstrate that mechanotransduction occurs in 3D networks. Although other features of network microarchitecture such as pore size may also impact cell behavior, the effect of these other features on cell behavior is outside the scope of our paper.

2) Even the 2D results must be carefully interpreted. For example, in a 2012 Nat. Mat. paper, some claim that it is not the matrix mechanical stiffness, but the number of collagen ligands. These issues are complex, and it is not clear cells are indeed sensing stiffness. I'm not sure how this paper adds to this discussion.

The *Nature Materials* paper referenced by the reviewer promotes the conclusion that stiffness matters, see Ref. 15. Specifically, these authors employed an approach that allowed them to modulate ECM stiffness independently from composition and architecture. They found that increasing ECM stiffness alone was enough to induce a malignant phenotype in normal mammary epithelial cells. Our paper has provided a fundamental, physical understanding of ECM stiffness and how it depends on the network properties. Although the overall behavior of a cell depends on a number of complicated factors, the mechanics of ECM are certainly one relevant factor in 3D. We have completely characterized the local stiffness of ECM and how it depends on network parameters, which is a necessary first step towards understanding cell behavior. To put our work in the proper biological context, we now cite Ref. 15 in the introduction of our paper.

3) The title of the paper is confusing at least to me. I am not clear what the authors mean by 'physical limits' to mechanical sensing. Is it because the local

stiffness is variable, it presents a limit to how the cells can sense the ECM stiffness? If so, it does not appear that this paper addresses this issue. The variation in local stiffness is to be expected, given the heterogeneous nature of fiber networks.

We have characterized the heterogeneity of disordered fiber networks by quantifying their mechanical response to an idealized measurement device. The resulting local stiffness distribution sets the uncertainty of stiffness inference, as we discussed in the last section of the Results in our main text. Since the limit is determined by the physics of disordered fiber networks rather than by the biological details of the cell's internal machinery, we have described our result as a "physical limit."

In the introduction to our paper, we noted that the variation in local stiffness is to be expected, given the heterogeneous nature of fiber networks. However, the universally large value we observed for the width of the local stiffness distribution is surprising and unexpected. Although we found that this effect is robust for biological networks over a range of parameters in multiple elastic regimes, we also observed that the width is narrower for disordered lattice networks, which had previously served as a standard model for fiber networks.

However, we do agree that the initial title was not focused enough. Therefore, we decided to change the title to "Physical limits to biomechanical sensing in disordered fiber networks."

4) As I mentioned, network moduli heterogeneity and how it depends on fiber structure and filament bending properties have been studied. I am not sure how this paper adds to this area.

We are not aware of any other work that elucidates the origin of the broadly heterogeneous local response we observed. To our knowledge, we have provided the first complete characterization how the local stiffness distribution depends on network properties, including fiber elasticity, the average coordination number, mesh size, and prestrain.

There is also a large field that work on constitutive modeling, using very sophisticated models that incorporate fiber geometrical properties and fiber mechanics. The actual problem is even more complicated, since the fiber themselves have heterogeneous bending properties, and variable bundle sizes. All of the parameters in Eq. S2 should have some distribution. I suspect real network exhibit even higher variable local stiffness.

We focused on a simplified model of fiber properties to elucidate the physical origin of mechanical heterogeneity in fiber networks. To validate these simplifications, we tested alternative mechanical models that incorporate a distribution of bending moduli, as well as different types of interactions at junctions. We did not observe a significant difference in modeled stiffnesses.

These results, taken together with our scaling model, suggest that the largest contribution to mechanical heterogeneity is due to structural network disorder, which is captured by our RGG model. Thus, we believe that our model provides a strong lower bound to the mechanical heterogeneity of a fiber network in the ECM. We have included results on these important clarifications in Supplementary Note 3.

5) The phase transition in Fig. 2 as a function of connectivity does seem interesting. Perhaps a theory can be developed there. But people working in the percolation area may already know about this.

In the case of macroscopic response, the analogy to percolation has been well characterized (see, e.g. Ref. 25), which is why we only briefly reviewed and discussed this topic in our main text.

6) The stiffness as a function of probe length does seem interesting. Here I would like to see more results. For example, as one varies collagen concentration, how does the effective network properties (Fig. 3) change?

We thank the referee for raising this important issue. We have expanded the discussion of the local stiffness as a function of probe length that appears in Supplementary Note 5. We have studied the mean stiffness versus probe length (previously Fig. 3) as a function of network connectivity, and found that the fold-stiffening can increase dramatically as a function of the proximity to an elastic transition.

7) When a different collagen or fibrin concentration is used to make the ECM, how do the results in the paper vary?

This is indeed a very important question. We performed additional experimental work to understand how our results vary as the concentration is increased. We found that higher concentrations of fibers result in denser networks of thicker fibers. Our results are consistent with an overall size-rescaling of the networks. Interestingly, upon applying the same size-rescaling to the probe length, the widths of the local stiffness distributions collapse onto universal values for different concentrations. We included a discussion of these additional findings in the first sections of the Results in the main text.

There are also issues like how ECM cross linking is done experimentally. I would have like to see a little more studies on network properties as a function of fiber density.

Since our experimental technique cannot access the mechanical behavior of the junctions, we assumed that all junctions are rigid in our initial mechanical model. However, some types of crosslinkers allow fibers to rotate or slide freely. We tested an alternative model for crosslinking and found that choosing

freely-hinging crosslinks at fourfold vertices (i.e. wherever two fibers may be crosslinked) does not significantly change our results. These additional modeling results are now included in Supplementary Note 3.

Reviewer #1 (Remarks to the Author):

I am satisfied with the response of the authors to my comments and revisions made by them in the manuscript. I recommend publication.

Reviewer #2 (Remarks to the Author):

The authors convincingly addressed all my previous comments. This paper can be accepted in its current form.

Reviewer #3 (Remarks to the Author):

In the revised paper, the authors did not address any of my critical comments at all. I remain unconvinced about the novelty and importance of the conclusions of the work. There is no attempt to verify any of the modeling conclusions, and the central point of the paper, namely there is heterogeneous local mechanical properties in the matrix is neither novel or convincingly demonstrated.

1) There are many uncertainties in the modeling approach, and to draw conclusions about real ECM requires much more scrutiny. For instance, collagen is a hierarchical structure with fibers of different diameter and composition. There is also polarity and electrostatic effects. Even without these basic considerations that many others have address in collagen matrices, the biggest unknown is the reference (undeformed) configuration that the modeling work assumes throughout. How is this reference configuration determined? As far as I see, it's chosen arbitrarily based on a lattice. There is no reason to take this assumption, and conclusions of the model will depend heavily on this assumption. This is why I stated in the previous comments that the real matrix is probably a lot more heterogeneous than what's described here. The authors dismissed this out of hand without in depth thinking.

2) My statement about mechanosensing itself and previous finding from the Nat. Materials paper also requires more thinking. The main message of that paper is that the cell is likely sensing ligand concentration instead of matrix mechanics. One cannot easily disentangle ligand concentration effects from the stiffness. Perhaps matrix mechanics is not important in real systems. The paper takes that at face value, and I think confuses these issues.

3) Broedersz and Mackintosh has written extensively about networks like this. This is another example of semi flexible networks that has been studied so well by this group. But here is no discussion of thermal effects, which in several regimes dominate the matrix elasticity. Why is not neglected here. Also please see my comments about model assumptions itself. It's likely that all parameters assumed at the filament level are also random. We don't understand their randomness.

Response to reviewer's comments:

Reviewer #3 (Remarks to the Author):

In the revised paper, the authors did not address any of my critical comments at all. I remain unconvinced about the novelty and importance of the conclusions of the work. There is no attempt to verify any of the modeling conclusions, and the central point of the paper, namely there is heterogeneous local mechanical properties in the matrix is neither novel or convincingly demonstrated.

1) There are many uncertainties in the modeling approach, and to draw conclusions about real ECM requires much more scrutiny. For instance, collagen is a hierarchical structure with fibers of different diameter and composition. There is also polarity and electrostatic effects.

The reviewer again raises the same legitimate concerns regarding the complexity of real ECM, which may contain fibers of varying diameter and concentration as well as additional extracellular components such as crosslinking proteins and proteoglycans. However, we believe that these concerns were all fully addressed in the revised version of our manuscript. In particular, the initial comments by the reviewers spurred additional modeling work that led to several new results and major changes throughout our manuscript. These major changes reported additional work performed to determine whether our results are robust to the presence of additional features found in real ECM. Specifically, in the main text, we considered the effects of varying fiber concentration (see page 2 and Fig. 3). By experimentally varying the concentration of fibrin, we found that the local stiffness distributions collapsed onto a universal form, which demonstrates that our results are robust to varying fiber composition. Next, thanks to the reviewers' helpful comments on our initial submission, we considered the role of preexisting stress, such as may be provided by proteoglycans. As noted by Reviewer 3, these proteoglycans are known to provide a combination of electrostatic and elastic effects that could impact the mechanical response (see page 6 of main text and Fig. 5). These additional considerations led to interesting new findings regarding the role of prestress. That is, we found that preexisting network stress could induce a transition from a bending-dominated to a stretching-dominated regime. We were able to account for this using our theoretical framework describing the bending-to-stretching transition. This additional successful application of our modeling framework demonstrates that our results are robust to varying network properties. Finally, in Supplementary Note 3 we discussed results on alternative mechanical models that account for the effects of varying fiber radii and the composition of crosslinkers at fiber junctions (see pages 6-8 of Supplementary Notes and Supplementary Fig. 3). For these alternative mechanical models, our results for the local stiffness measurements were largely unchanged on a case-by-case

basis. Taken together, the additional modeling work we performed for our first revision confirms that our results are robust to the effects brought up by the reviewer and should therefore apply to real ECM.

Even without these basic considerations that many others have address in collagen matrices, the biggest unknown is the reference (undeformed) configuration that the modeling work assumes throughout. How is this reference configuration determined? As far as I see, it's chosen arbitrarily based on a lattice. There is no reason to take this assumption, and conclusions of the model will depend heavily on this assumption. This is why I stated in the previous comments that the real matrix is probably a lot more heterogeneous than what's described here. The authors dismissed this out of hand without in depth thinking.

Throughout our analysis, we considered many types of reference configurations, both experimental (collagen and fibrin) and theoretical (the off-lattice random geometric graph model and FCC lattice models). It is precisely the universality of our results for all these types of networks that underpins the importance of our results. We certainly did not dismiss the reviewers remarks, rather we have consistently addressed these concerns in every version of the manuscript. In our initial submission, we addressed network heterogeneity by considering an off-lattice model, i.e. the "random graph" network, as well as experimentally-determined networks. Furthermore, in the previous revised manuscript, for the experimental collagen network, we considered both undeformed reference states as well as deformed reference states that could arise from the presence of proteoglycans or cells (see response above). By comparing the random graph networks to the lattice networks, we found that the presence of long fibers can broaden the distribution. This broadening confirms the reviewer's intuition about the additional heterogeneity of real ECM, and also results in a local stiffness distribution that closely matches the experimental data.

2) My statement about mechanosensing itself and previous finding from the Nat. Materials paper also requires more thinking. The main message of that paper is that the cell is likely sensing ligand concentration instead of matrix mechanics. One cannot easily disentangle ligand concentration effects from the stiffness. Perhaps matrix mechanics is not important in real systems. The paper takes that at face value, and I think confuses these issues.

The 2014 Nature Materials paper by Chaudhuri et al. (Ref. 15) provides clear evidence for the importance of matrix stiffness as a relevant factor that influences cell behavior. This interpretation is supported by mechanosensing researcher, Prof. Sanjay Kumar, who summarized the 2014 paper in a News and Views piece for Nature Materials titled "Cellular mechanotransduction: Stiffness does matter." The following is a highly relevant excerpt from this piece:

“...now, two studies published in Nature Materials explore these potential confounding effects. On the one hand, David Mooney and colleagues decoupled ECM stiffness from ligand density by creating an interpenetrating polymer network (IPN) system, and used it to dissect how stiffness controls mammary epithelial morphogenesis in three-dimensional culture. On the other hand, Adam Engler and colleagues systematically modulated the stiffness, porosity and ligand-substrate coupling (the latter two affect ligand tethering) of polyacrylamide (PAAm) gel culture substrates, and studied how all these properties affect the differentiation of tissue stem cells. These studies show that, in practice, stiffness rather than ligand tethering governs mechanosensitive cell adhesion and differentiation...”

This summary article concludes that both stiffness and ligand density can influence the response of a cell. Our work provides the first theoretical framework to understand the role of matrix stiffness on mechanosensing, which is a crucial first step in determining the complete behavioral response of a cell to its mechanical environment. Any studies performed on ligand density in the future will be complemented and clarified by our results.

3) Broedersz and Mackintosh has written extensively about networks like this. This is another example of semi flexible networks that has been studied so well by this group. But here is no discussion of thermal effects, which in several regimes dominate the matrix elasticity. Why is not neglected here. Also please see my comments about model assumptions itself. It's likely that all parameters assumed at the filament level are also random. We don't understand their randomness.

We thank the reviewer for pointing out the lack of discussion of thermal effects, and we have modified our manuscript to include this discussion. Thermal effects may arise from bending fluctuations, or alternatively, from entropic contributions to the longitudinal compliance of fibers that are stiff to bending fluctuations. We neglected bending fluctuations because the persistence lengths of polymers comprising extracellular fiber networks are expected to be well above the corresponding mesh size of the network. To validate this assumption, we have estimated the average persistence lengths for the experimental networks we considered. We found that for all types of experimental networks considered, the average persistence length is orders of magnitude larger than the mesh size. We therefore expect that the fibers bend as semiflexible rods on such scales. Next, we estimated the relative contributions of mechanical and entropic contributions to the stretching modulus, and found that the mechanical contributions dominate. We note that this assumption is not critical for bending-dominated networks, a regime in which the stretching modulus does not contribute to the mechanical response. Since biopolymers are much softer to bending than to stretching, we consider that our experimental networks are bending-dominated in linear response and

thus our main results do not receive contributions from thermal effects. Taken together, our new estimates confirm that we can safely neglect any thermal effects. The new calculations are now presented in Supplementary Note 3 (see page 5). Regarding the model assumptions, we thoroughly addressed the randomness in the filament composition in the previous revised manuscript (see response to point 1). In particular, based on the experimentally observed variability of the fiber diameters (Ref. 51), we considered a variant of our mechanical model with a corresponding amount of randomness (see page 6 of Supplementary Note 3). We also considered an alternative model for the mechanics of fiber junctions (see page 6 of Supplementary Note 3). We found that for both of these additional effects, our results were not significantly altered by the randomness in the parameters at the filament level.

Reviewers' comments:

Reviewer #3 (Remarks to the Author):

The authors have answered most of my questions. I still maintain that the actual local stiffness is significantly more random than what's obtained, for reasons that have not been examined by this model. Why only vary the fiber diameter from 25-500nm? Depending on the tissue type, the fibers can be microns in size. In addition connectivity is not absolute, and there would be significant plastic deformation as well. So effectively I think the matrix a lot more heterogeneous. I think the authors should state this, and not limit to just bending modulus variation.

Also, I find it a bit strange that the collagen I concentration ranges from 0.2-1.8mg/ml. The typical quoted physiological concentration goes from 2mg/ml to above. Anything lower than 2mg/ml, fibroblasts actually fall out of the matrix. There wouldn't be an intact matrix from the point of view of the cell. From our experience, 2mg/ml is hardly a gel at all, its mostly a viscous liquid. Also, the method of gelation makes a huge difference in terms of mechanical properties. This suggest that crosslinking chemistry actually dominates the mechanics of the network. The type of idealized model here doesn't really capture the dominant contributions.

A minor point, Fig 2 caption in the SM is confusing. I think the authors labeled the panels incorrectly.

Response to reviewer's comments:

Reviewer #3 (Remarks to the Author):

The authors have answered most of my questions. I still maintain that the actual local stiffness is significantly more random than what's obtained, for reasons that have not been examined by this model.

We thank the Reviewer for drawing attention to the importance of clarifying this point, and we note that any additional randomness of the local stiffness would further strengthen the central claim of our paper regarding the fundamental importance of mechanical heterogeneity to cellular mechanosensing. Throughout the main text of our paper we focused on simplified models that capture the mechanical heterogeneity due to intrinsic structural disorder. As the Reviewer points out, biological ECM may contain additional sources of randomness. Therefore, our simplest models set a lower bound on the total heterogeneity. To confirm that this lower bound provides a good quantitative estimate for the heterogeneity of stiffness, as detailed in Supplementary Note 3 of our paper, we have tested alternative mechanical models that incorporate the most likely sources of additional heterogeneity. In the previous version of our Supplementary Notes, these additional sources of heterogeneity included a distribution of bending moduli and different types of interactions at junctions. Thanks to the useful suggestion from the Reviewer in these most recent remarks (see below), we now also include a variant model that allows for plastic deformations (detailed in Supplementary Note 3 starting on page 9). In all these additional studies, we observed no significant difference in the modeled local stiffness distribution for the experimental collagen network (Supplementary Figs 3 and 4). This robustness to additional randomness indicates that the large heterogeneity due to intrinsic structural disorder dominates over other sources of disorder, implying that our simplified models provide a relevant quantitative estimate for the total mechanical heterogeneity. We now include an additional paragraph in the Discussion of our main text to clarify this important feature of our results.

Why only vary the fiber diameter from 25-500nm? Depending on the tissue type, the fibers can be microns in size.

The types of networks used in our study, i.e. collagen and fibrin networks of concentrations around 2 mg/ml, have been imaged experimentally (see Ref. 52 and 53). The relevant range of fiber radii was found to be around 25-100 nm. Using this experimentally observed range, we showed that the intrinsic structural disorder of our collagen network dominates over the mechanical disorder due to variations in fiber radii (see section starting on page 6 of Supplementary Notes). Furthermore, although we have investigated multiple types of biopolymers and studied several models, it is not our aim to develop a specific detailed model for each type of tissue. Rather, we have identified

general principles that should be relevant for most tissues. We now include a reference to experimental observations of the fiber diameter distribution for fibrin (Ref. 53).

In addition connectivity is not absolute, and there would be significant plastic deformation as well. So effectively I think the matrix a lot more heterogeneous. I think the authors should state this, and not limit to just bending modulus variation.

We thank the Reviewer for providing helpful suggestions of additional sources of heterogeneity to investigate. Network connectivity is known to change significantly due to matrix remodeling that occurs on scales of hours to days (Ref. 45). We have noted in our Discussion that cells have been observed to perform measurements of stiffness that occur over minutes to hours. Thus, the long timescale dynamics due to connectivity remodeling can reasonably be neglected for our purposes. However, plastic deformations have been observed for large values of strain on a time scale of tens of seconds (Ref. 51). Here, direct observations using confocal microscopy revealed that the plasticity occurs due to persistent fiber lengthening, rather than changing the connectivity of the matrix. To understand the effect of this type of plasticity on the local stiffness distribution, we extended our model for nonlinear deformations to allow for irreversible fiber lengthening (Supplementary Note 3). We found that allowing fibers to irreversibly lengthen under large prestrain did not significantly alter the local stiffness distribution (Supplementary Fig. 4c). These observations are consistent with our previous results that suggest the width of the local stiffness distribution is dominated by intrinsic structural disorder rather than by variations in the mechanical properties of individual fibers and their interactions.

Also, I find it a bit strange that the collagen I concentration ranges from 0.2-1.8mg/ml. The typical quoted physiological concentration goes from 2mg/ml to above. Anything lower than 2mg/ml, fibroblasts actually fall out of the matrix. There wouldn't be an intact matrix from the point of view of the cell. From our experience, 2mg/ml is hardly a gel at all, its mostly a viscous liquid.

We have used macroscopic rheology to confirm that our networks behave as gels. Specifically, we observed that they have nonzero elastic moduli, which can only occur for intact, connected matrices. We also note that the concentrations of fibrin we used in our study are approximately equal to the concentrations observed for blood clots (Ref. 53). Furthermore, the concentrations of both types of biopolymers used in our study are similar to the values that are commonly used for experimental studies of cell migration, for example in the recent *Nature Communications* paper "Local 3D matrix microenvironment regulates cell migration through spatiotemporal dynamics of contractility-dependent adhesions" (referenced in the review article Ref.

10). In addition, we now also cite a different study in which the authors directly observed cells migrating through collagen networks of concentrations as low as 1 mg/ml (Ref. 36). Interestingly, the authors of that study found that several measures of cell motility (including persistence time and diffusivity) varied significantly for collagen concentrations of 1 mg/ml versus 2 mg/ml, which are similar to the range of concentrations explored in our work. Finally, we have systematically varied the concentration of fibrin and found that the local stiffness distribution takes on a universal form (see Results section of our manuscript), which suggests that our results also apply to higher concentration networks. We have modified the Discussion section accordingly, adding additional context concerning the experimental concentrations used in our study.

Also, the method of gelation makes a huge difference in terms of mechanical properties. This suggest that crosslinking chemistry actually dominates the mechanics of the network. The type of idealized model here doesn't really capture the dominant contributions.

Our model is able to capture the effect of crosslinking chemistry by varying the bending moduli of fibers, which determine the torsional stiffnesses of interactions at the fiber junctions. The possible effects of additional mechanical heterogeneity due to varying crosslink chemistry is addressed by the variant mechanical models that we studied in Supplementary Note 3 in the previous version of our paper. In contrast to the suggestion that crosslinking chemistry might dominate the mechanics, we observed that the local stiffness is dominated by the intrinsic structural disorder of the network rather than by mechanical heterogeneity due to varying junction stiffness or crosslink rigidity (Supplementary Fig. 3).

A minor point, Fig 2 caption in the SM is confusing. I think the authors labeled the panels incorrectly.

We thank the Reviewer for pointing out this error in the caption to Supplementary Fig. 2, which we have now corrected.

REVIEWERS' COMMENTS:

Reviewer #3 (Remarks to the Author):

The authors have addressed my comments, I recommend publication.